# Climate resilience of European wine regions

**Simon Tscholl** [1,2,6] ✉, **Sebastian Candiago** [1,3,4,6], **Thomas Marsoner** [1],
**Helder Fraga** [5], **Carlo Giupponi** [3] & **Lukas Egarter Vigl** [1]

Over centuries, European vintners have developed a profound knowledge about grapes, environment, and techniques that yield the most distinguishable wines. In many regions, this knowledge is reflected in the system of wine geographical indications (GI), but climate change is challenging this historical union. Here, we present a climate change vulnerability assessment of 1085 wine GIs across Europe and propose climate-resilient development pathways using an ensemble of biophysical and socioeconomic indicators. Results indicate that wine regions in Southern Europe are among the most vulnerable, with high levels also found in Eastern Europe. Vulnerability is influenced by the rigidity of the GI system, which restricts grape variety diversity and thus contributes to an increased sensitivity to climate change. Contextual deficiencies, such as limited socioeconomic resources, may further contribute to increased vulnerability. Building a climate-resilient wine sector will require rethinking the GI system by allowing innovation to compensate for the negative effects of climate change.

The concept of geographical indication (GI) plays an essential role in defining a wine's identity and establishing a strong link between the product's unique characteristics and its provenance[1]. Indeed, many of the world's most famous wines are known for their origin and not for their grape variety[2]. The system of classifying and regulating wines based on their origin is commonly referred to as Geographical Indication (GI)[3], and the strictest rules can be found in Europe, where premium GI wines are labelled as Protected Designation of Origin (PDO)[4]. These wines can only be produced in legally defined areas that have been selected based on soil type, climate, and historical or administrative divisions. The presence of both human and natural dimensions in defining wine regulations is related to the historic concept of Terroir: an originally French notion that states that the place (both the land with its climate and the people) defines the product[5].

Climate change is increasingly impacting several aspects of viticulture, including vine phenology[6–8], grape composition[9–11] and growing suitability[12–15]. These biophysical changes require growers and producers to adapt by employing new cultivation techniques, using new varieties, or shifting cultivation locations[16–19]. However, the legal rigidity of the GI system can impair the ability of wine regions to adapt and to preserve traditional wine production in the context of climate change[20], i.e., GI resilience. For example, Burgundy and Champagne are known for wines made from *Pinot Noir*. If these regions become unable to grow typical *Pinot Noir* grapes at some point, they are under serious threat. A substitute vine variety would neither qualify for the label, nor would the law permit growers to source grapes from outside the region or introduce new cultivation techniques[21,22] without going through the process of amending the wine region's regulations[23]. In many wine regions, increasing resilience will, therefore, depend upon adaptation strategies that overcome traditional and legislated practices by including more flexibility to better support the sustainable development of winemaking in uncertain climates.

Assessing the vulnerability of wine GIs to climate change facilitates the understanding of which regions are threatened the most by climate change and supports the development of potential adaptation

[1]Institute for Alpine Environment, Eurac Research, Viale Druso 1, 39100 Bozen/Bolzano, Italy. [2]Department of Ecology, University of Innsbruck, Innrain 52, 6020 Innsbruck, Austria. [3]Department of Economics, Ca' Foscari University of Venice, S. Giobbe 873, 30121 Venezia, Italy. [4]Professorship of Ecological Services, Bayreuth Center of Ecology and Environmental Research (BayCEER), University of Bayreuth, Universitätsstraße 30, 95447 Bayreuth, Germany. [5]Centre for the Research and Technology of Agro-Environmental and Biological Sciences (CITAB), Institute for Innovation, Capacity Building and Sustainability of Agri-food Production (Inov4Agro), Universidade de Trás-os-Montes e Alto Douro (UTAD), 5000-801 Vila Real, Portugal. [6]These authors contributed equally: Simon Tscholl, Sebastian Candiago. ✉e-mail: simon.tscholl@eurac.edu

pathways to strengthen their resilience. The vulnerability depends on the individual characteristics of each wine region, including the degree of climate exposure and sensitivity and the availability of socioeconomic and biophysical resources, which strongly determine how wine GIs can adapt to climate change[24]. The importance of exposure and sensitivity has already been extensively investigated, for instance, by relating changes in air temperature or precipitation to relevant vine parameters[7,14,15,25,26], or analyzing how grapevine diversity and variety turnover influence future land suitability[12,14,27,28]. Although vulnerability assessments have been used in other sectors[29,30] and for other crops[31–33], in the context of wine GIs they have been sparse and thus far, limited to single wine regions[34–38]. Additionally, the focus of previous studies on climate change adaptation was primarily on bioclimatic pressures, while the legal and socioeconomic parts have often been neglected[39]. The future of the GI system under climate change is, therefore, still poorly understood, and our knowledge of how adaptive capacity and climate change vulnerability are related to the resilience of wine GIs is very limited.

In this study, we assess the climate change vulnerability of 1085 wine regions in Europe, which all produce wines labelled as PDO, by explicitly considering their biophysical and socioeconomic characteristics and their regulatory specifications. We use a recently published dataset on European wine GIs[40] coupled with an index-based approach including an ensemble of financial, biophysical, and social indicators. To assess climate change vulnerability, we adopt the framework developed by the Intergovernmental Panel on Climate Change (IPCC) and calculate an integrated vulnerability index considering exposure, sensitivity, and adaptive capacity[24]. We define (i) exposure as the expected changes in bioclimatic conditions important for viticulture, (ii) sensitivity as the degree to which PDO regions are affected by climate-related stimuli, based on the historical growing conditions of the typical vine varieties of each wine region (bioregional

climate range), and (iii) adaptive capacity as the potential of a wine region to adapt to changing climate conditions, considering five distinct dimensions (financial, natural, physical, social, and human) (see Methods). We carry out a comparative analysis that provides a basis for the discussion of possible future pathways related to the climate resilience and adaptation of the GI system. As such, the results represent a first step in assessing the impact of climate change on designated wine GIs across Europe and can be used to identify critical elements that should be the focus of future research.

## Results

### Exposure and sensitivity to climate change

Climate variability has always affected winemaking, but the current rate of climate change is unprecedented, challenging the historical union between favorable site conditions, optimum grape varieties, and traditional viticultural practices. We defined exposure as the degree to which climate is projected to change in wine regions, with 0 indicating regions with the lowest exposure and 1 regions with the highest exposure. The highest levels of exposure were observed in Romania, Croatia, Bulgaria, Italy, and Hungary, with an average exposure level above 0.7 (Fig. 1a). Many of these regions are located in mountainous terrain, especially in the Apennines, Alps and Carpathian Mountains. In contrast, lower levels of exposure were found in areas with a strong oceanic influence on climate, such as Portugal or the Canary Islands, as well as at higher latitudes, such as in Belgium, and the Netherlands, with an average exposure level below 0.4. In general, there is a trend toward increased temperatures in most regions, leading to an increased Huglin Index and Cool Night Index, combined with drier conditions, as indicated by the decrease of the Dryness Index (Fig. 1c). The observed trends are consistent with other studies that use CMIP-6 scenarios[24,41]. Our results are also in line with other studies analyzing climate change impacts on European viticulture, many of which

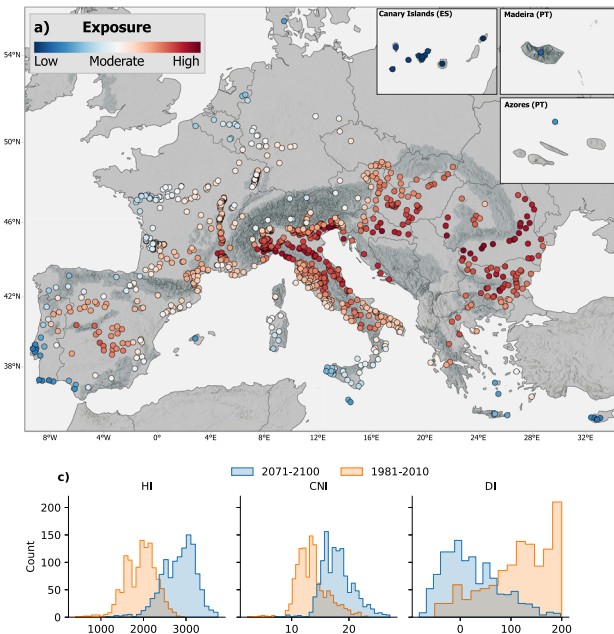

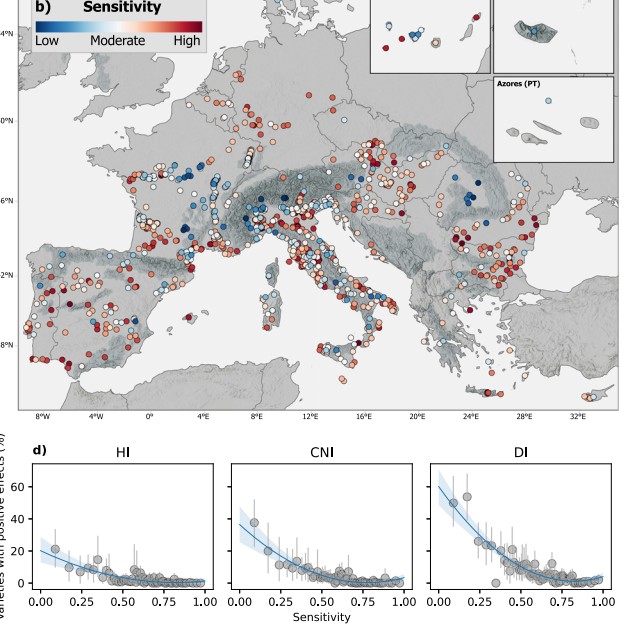

**Fig. 1 | Exposure and sensitivity to climate change of European wine GIs.**
**a, b** Map of climate change exposure and sensitivity of the European wine GIs. The regions are indicated by their geographical center. Dark gray areas in the background represent mountain regions. Made with Natural Earth. Free vector and raster map data @ naturalearthdata.com. **c** Histogram showing the distribution of HI, CNI and DI for the periods 1981–2010 and 2071–2100 under the ssp370 scenario of all wine regions. **d** The sensitivity of European wine GIs related to the share of varieties with potential positive effects from climate change (see methods) for each

PDO under the ssp370 scenario and for the period 2041–2070 considering three bioclimatic indices. The blue line shows a polynomial model between the two variables. Points for each x-value have been aggregated using the mean function to reduce the number of points in the plot. Each point represents $n = 18$ regions (total number of regions = 1085). Gray bars around each point indicate the 95% confidence interval. HI Huglin Index, CNI Cool Night Index, DI Dryness Index, ES Spain, PT Portugal.

**Table 1 | Indicators of adaptive capacity**

| Dimension | Indicator | Description | Unit |
|---|---|---|---|
| Social | Aging index | Ratio between old and young population | — |
| | Dependency ratio | Ratio between the dependent and working population | — |
| | Population density | Population density per agricultural area | n/ha |
| Physical | Road length | Total length of roads potentially usable for viticulture | km |
| | Mechanization Index | Machinery & equipment in use per vineyard area | €/ha |
| | Naturalness | Share of natural and semi-natural areas in winegrowing areas | % |
| Natural | Shift in space | Available areas with cool climatic conditions suitable for viticulture | km² |
| | Water availability | Excess water from precipitation available in winegrowing areas | mm |
| | Availability of climatic niches | Spatial variability of thermal conditions within a winegrowing region | °C |
| Human | Labor force | Percentage of regular from total farm labor force | % |
| | Education level | Education level of farm managers | — |
| | Research accessibility | Proximity to closest research center on wine and vine | km |
| Financial | Debt ratio | Liability percentage of total assets of wine farms | % |
| | Return on assets | Profitability in relation to total assets of wine farms | % |
| | Subsidy dependence | Net income percentage coming from subsidies of wine farms | % |

observed high levels of impacts in areas that correspond to high-exposure regions in our study. For instance, strong yield decreases were projected for northern Italy, central Spain, Greece, and Bulgaria[7] and decreased suitability for Spain, parts of France, central and northern Italy, and large parts of eastern Europe[14]. To account for the potential uncertainties in future climate predictions, we also analyzed the model spread among the 5 GCMs used in the present study for both temperature and precipitation over Europe (Supplementary Figs. 5 and 6). Stronger differences in the predictions of individual models indicate a higher uncertainty for future climate predictions[42]. Temperature differences between the 5 models increase towards the end of the century for more pessimistic scenarios, especially in some Central and Eastern European regions, indicating a higher model uncertainty for these areas. In contrast, precipitation differences between the 5 GCMs are highest in mountainous regions, such as the European Alps, but remain similar across different time periods.

Climate change is also altering the traditional identity of GIs by shifting climatic conditions either closer to the climate optimum or pushing them further away. As such, the sensitivity level describes the degree to which viticulture in a certain region is affected by climate-related stimuli based on the climate ranges of traditionally cultivated varieties. Typical vine varieties are particularly important for wine PDOs because the regulatory documents that protect the name of specific products and promote their unique characteristics are mostly organized around specific wine products with their associated varieties. We found that regions in southern Europe often tended to have higher sensitivity levels either due to a limited grape variety spectrum or due to warm climatic conditions close to the upper limit of their varietal ranges (Fig. 1b). However, we also found regions with low sensitivity levels in Southern Europe, e.g., Do Tejo (PT), and regions with increased sensitivity at higher latitudes, e.g., Champagne (FR). For regions with a low sensitivity, where current climatic conditions are comparatively far below the upper threshold for certain varieties, some authorized varieties might experience positive effects from climate change (Fig. 1d). However, these positive effects strongly depend on the magnitude of climate change and severely decline under stronger changes in climatic conditions (Supplementary Fig. 7). Thus, if climate change is to proceed at the current rate, most of the GIs as we know them now will necessarily change because the best location for a given variety today might be the best location for a different variety in the future[43]. The diversity of cultivated varieties will therefore, be a critical factor in determining the magnitude of future impacts[27].

The lack of variety-specific information, including their spatial distribution and phenological characteristics, poses a significant limitation to our capacity to estimate future impacts of climate change in GIs. For instance, there is a very large diversity of phenological characteristics amongst different vine varieties[44], but detailed, variety-specific knowledge is mostly limited to international varieties representing approximately 1% of global vine diversity[27]. This only covers a very small proportion of the more than 1000 varieties currently listed for the European PDO regions. In the present approach, we therefore used the regulatory information of the GIs coupled with their spatial distribution to derive an estimation of the bioregional climate range for each variety. However, specific information for a broader range of varieties would allow a more thorough assessment of the climate change sensitivity, for instance, by using phenological models or developing distribution models for individual varieties under different scenarios[7,27,45], and thus a more detailed analysis of the climate change vulnerability.

## Adaptive capacity to climate change

To adapt and cope with climate change, GI regions need access to resources which enable and facilitate the implementation of adaptation strategies. The adaptive capacity indicates the readiness and potential of a region to adapt to climate change. To assess the adaptive capacity of the European wine regions, we identified five crucial dimensions which characterize the ability of wine regions to counteract the negative effects of climate change[46,47]: (i) the financial dimension, describing the financial situation of farms specialized in viticulture in a region; ii) the natural dimension, reflecting the topoclimatic diversity within a region; (iii) the physical dimension, describing the presence of infrastructure and physical assets for viticulture; and the (iv) social and (v) human dimensions, which are related to population characteristics, such as age structure or employment, and education as well as available labor force within the wine regions (Supplementary Figs. 8–12). We considered a total of 15 indicators of adaptive capacity that together estimate the potential of wine regions to adapt to climate change (Table 1). For more detailed information on the considered indicators, the calculation method, unit, and the rationale behind their inclusion, please refer to the methods section as well as Supplementary Note 1.

Some European wine regions with the highest adaptive capacity were found within or near the European Alps and along the Apennines (i.e., on the west coast of the Italian peninsula) (Fig. 2a), for example, Conegliano Valdobbiadene Prosecco (IT) and Alto Adige (IT) (Fig. 2b).

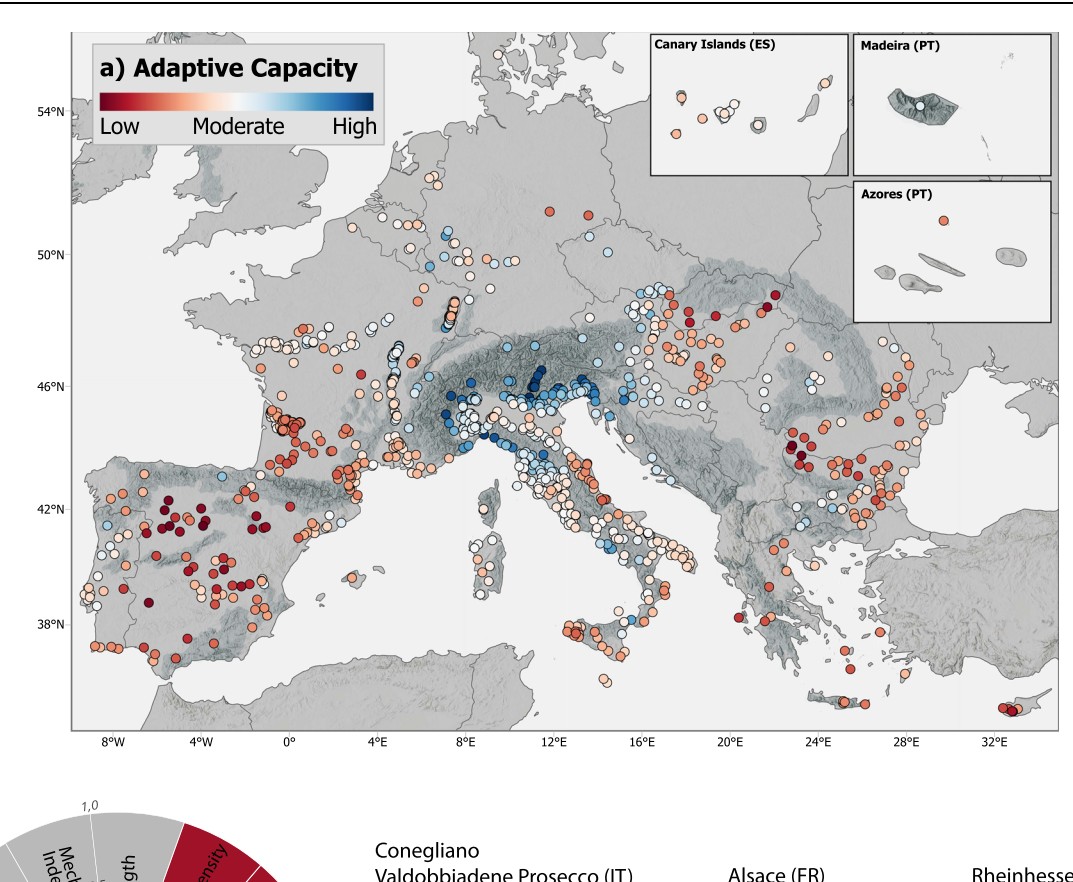

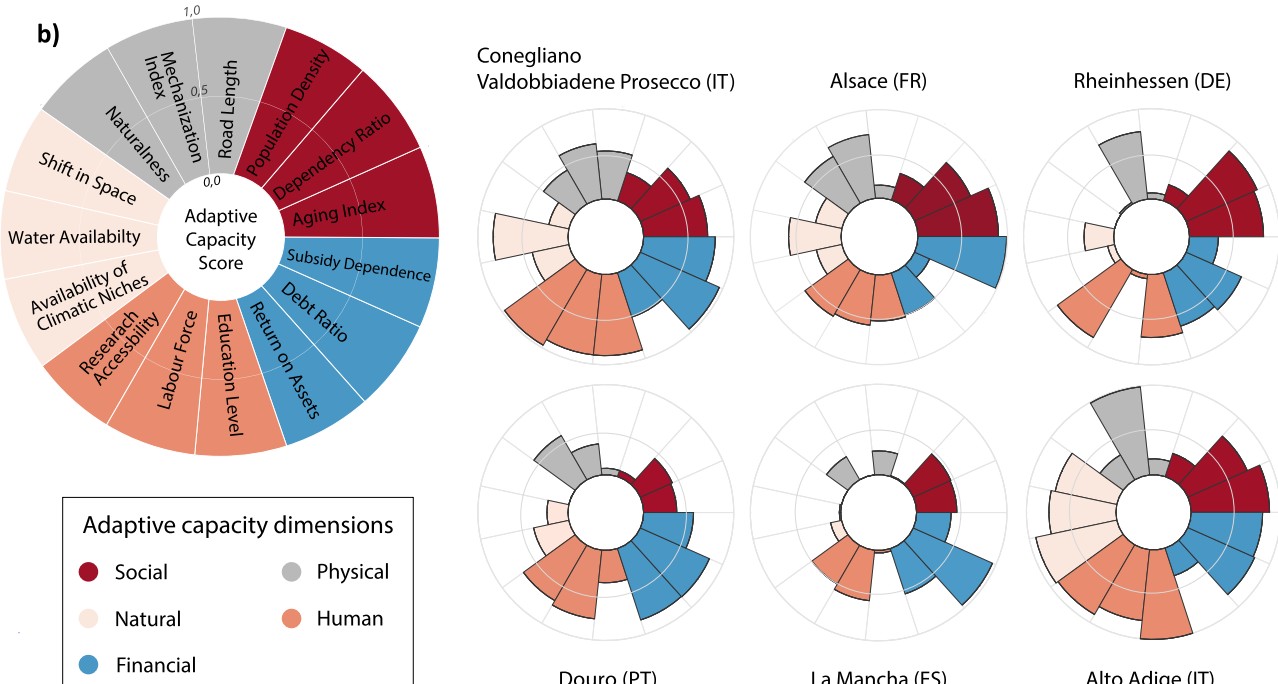

**Fig. 2 | Adaptive capacity of the European wine GIs. a** Map showing the adaptive capacity of European wine GIs. The points refer to the centroids of the regions. Dark gray areas refer to mountain regions. Made with Natural Earth. Free vector and raster map data @ naturalearthdata.com. **b** Petal diagram showing the five dimensions (colors) of adaptive capacity and the related indicators for six selected GIs. IT Italy, FR France, DE Germany, PT Portugal, ES Spain.

Slovenia and Italy were the countries with the highest share of regions with an adaptive capacity level in the upper quartile (65% and 14%, respectively), followed by France with less than 10%. In contrast, regions in central Spain and eastern Europe, such as Slovakia, Greece, Romania, Bulgaria, and Hungary, had low adaptive capacity levels with average values below 0.3. The regions in Spain tended to have a high financial capacity; however, they had low scores for all the other dimensions, especially the physical and natural capacity, resulting in a low overall adaptive capacity (e.g., La Mancha (Fig. 2b)). Winegrowing regions at higher latitudes, including some regions in France, Germany, Denmark, Belgium, or the Netherlands, mostly had moderate adaptive capacity levels around 0.5 (e.g., Rheinhessen (DE) and Alsace (FR) (Fig. 2b)).

Despite the growing evidence that adaptive capacity plays a central role in climate change adaptation and vulnerability[48,49], local-scale data for the calculation of adaptive capacity in viticulture is not yet

available in Europe. In addition, the extent to which individual indicators contribute to overall adaptive capacity is largely unknown but can vary significantly depending on the region's characteristics. (see Supplementary Fig. 13). As there is currently no local-scale data available for viticulture across Europe, including information on the type, scale, management practices, and market orientation, it is not possible to derive region-specific weights at a pan-European level. Our assessment, therefore, assumes equal weight in the importance of the 15 indicators constituting the adaptive capacity. The results of our calculation need to be contextualized in the framework of a continental scale analysis that delivers a high-level comparison of the adaptive capacity levels. However, this analysis cannot fully represent the individual differences and needs of each region's adaptive capacity.

### Climate-resilient winegrowing

The integrated vulnerability index of the wine regions is determined by the combination of their exposure, sensitivity, and adaptive capacity and indicates their susceptibility to the adverse impacts of climate change (Fig. 3). By analyzing the combined effect of these three indicators, it is possible to identify groups of regions with similar characteristics and give insights into potential pathways towards a more climate-resilient wine sector.

Five percent of European wine regions have a very high integrated vulnerability index (Group 6). They are likely to face the strongest negative impacts within the next few decades, due to their high levels of exposure and sensitivity, and their limited resources for adaptation[7,14,15,50] (Fig. 3b). Examples from this group were located in Bulgaria (e.g., Southern Black Sea), Romania (e.g., Oltina), Hungary (e.g., Hajós-Baja), parts of Italy (e.g., wine regions of Trebbiano d'Abruzzo and Lambrusco Mantovano) and Spain (e.g., Sierra de Salamanca). Some of these regions have already started to employ specific climate change adaptation strategies in their production regulations, such as redefining some categories of wine products (e.g., Kunság (HU)), or updating the analytical parameters of wines (e.g., Cebreros (ES)). Ultimately, however, the capability of these highly vulnerable regions to face climate change is restricted by their limited adaptive resources.

Groups 3, 4 and 5 represent regions with a high integrated vulnerability index that are in a better position compared to the regions in Group 6 but might still experience severe impacts from climate change. Examples include regions in southeast France (e.g., Côtes de Provence), northern Italy (e.g., Conegliano Valdobbiadene Prosecco), Slovakia (e.g., East Slovak), the Iberian Peninsula (e.g., Alentejo (PT) and Rioja (ES)) and some regions in the Apennines. The regions in these groups are highly heterogeneous in terms of the composition of exposure, sensitivity, and adaptive capacity. For instance, regions in Group 3 will likely face very strong negative impacts from climate change due to their high exposure and sensitivity. However, they also have a comparatively high adaptive capacity and are, therefore, better able to adapt to these impacts than regions with less available resources. In contrast, regions in Group 4 have a lower sensitivity but also fewer available resources for adaptation, while regions in Group 5 have a comparatively low exposure combined with high sensitivity and low adaptive capacity. Overall, the individual characteristics of each group, such as the varietal diversity, the availability of resources for adaptation or the magnitude of expected changes in climatic conditions, will determine the future development and the climate resilience of these wine regions.

Regions with low and medium integrated vulnerability index levels are represented by Group 1 and 2, respectively. Although some of these regions will face significant changes in climate, their higher adaptive capacity or lower sensitivity reduces their vulnerability to such adverse effects. This gives them the best prospects of maintaining their historic identity and high production standards in the medium to long term. These clusters include some regions at higher latitudes

(e.g., Rheinhessen (DE) and Crémant de Wallonie (BE)), in France (e.g., Côtes d'Auvergne and Alsace), or the European Alps (e.g., Alto Adige (IT)). Regions with a low sensitivity, located often at higher latitudes or within mountain regions, might even benefit from climate change to a certain degree[51–53]. However, the number of varieties with potential benefits from climate change decreases strongly under more pessimistic climate change scenarios[45] (Supplementary Fig. 7). In the context of GIs, this decline is particularly strong because the expected positive effects from climate change are often limited by the restricted range of varieties that are authorized to be grown. Even PDO regions located in comparatively cool climatic conditions might, therefore, not directly benefit from climate change in the long term and under moderate to high emission scenarios without changing their production regulations.

Vulnerability is a central concept in climate change studies that has been conceptualized in a multitude of different ways and contexts, mainly depending on the scope and scale of the analysis[54]. The case study of European PDO regions presented here provides a clear example of how the different facets of climate change can be combined into a single assessment, with its associated methodological constraints. The proposed integrated vulnerability index works well in the context of a comparative analysis, as it allows the ranking of different entities based on several variables. Our results should, therefore, be seen as a comparative overview of the climate change vulnerability across European winegrowing regions that can be used to distinguish critical, highly vulnerable regions from those less threatened by climate change. However, the approach can also be sensitive to the strategy used to combine the three dimensions of vulnerability (e.g. additive vs multiplicative data aggregation) and their respective weights. The results should, therefore, be seen in this context and as a first step toward a better understanding of the climate change vulnerability of viticulture in Europe.

## Discussion

Adaptation strategies generally need to be chosen on a region-specific basis, as they depend on the individual characteristics of each region. In general, however, most GIs will aim to maintain the identity and characteristics of the wines for which they are known today also under future conditions. Based on the integrated vulnerability index of the European wine regions combined with results from previous studies, it is therefore possible to outline some potential future pathways for the European wine sector. These can help to identify sound adaptation options and increase the resilience of the GI system.

For instance, adjustment of viticultural processes has been shown to compensate for a change in climatic conditions up to a certain degree[55] and might thus be useful for regions with high exposure levels. This includes strategies such as canopy management, use of irrigation, modification of vineyard structure, selection of rootstocks or the use of cover crops. They can be used to modify the interactions between vines and the environment (e.g., uptake of water and nutrients) or to directly alter microclimatic conditions in vineyards (e.g., exposure to incoming radiation) and thus can reduce negative effects from climate change. In the case of high sensitivity levels, adjusting wine blend ratios and compositions or gradually introducing new varieties could be promising options to mitigate potential changes in grape composition and wine style[27,43]. This is, for example, currently occurring in the region of Bordeaux, where new varieties have been added to the production regulations and are being tested extensively[56]. The strongest climate impacts will be found in regions with both high exposure and sensitivity, which may lead to significant shifts in product characteristics. In extreme cases, adaptation measures that go beyond vineyard management, such as site relocation to new climatic conditions, may be necessary[17]. However, the scope of such adaptation measures often entails a partial or complete departure from traditional production regulations and might also strongly

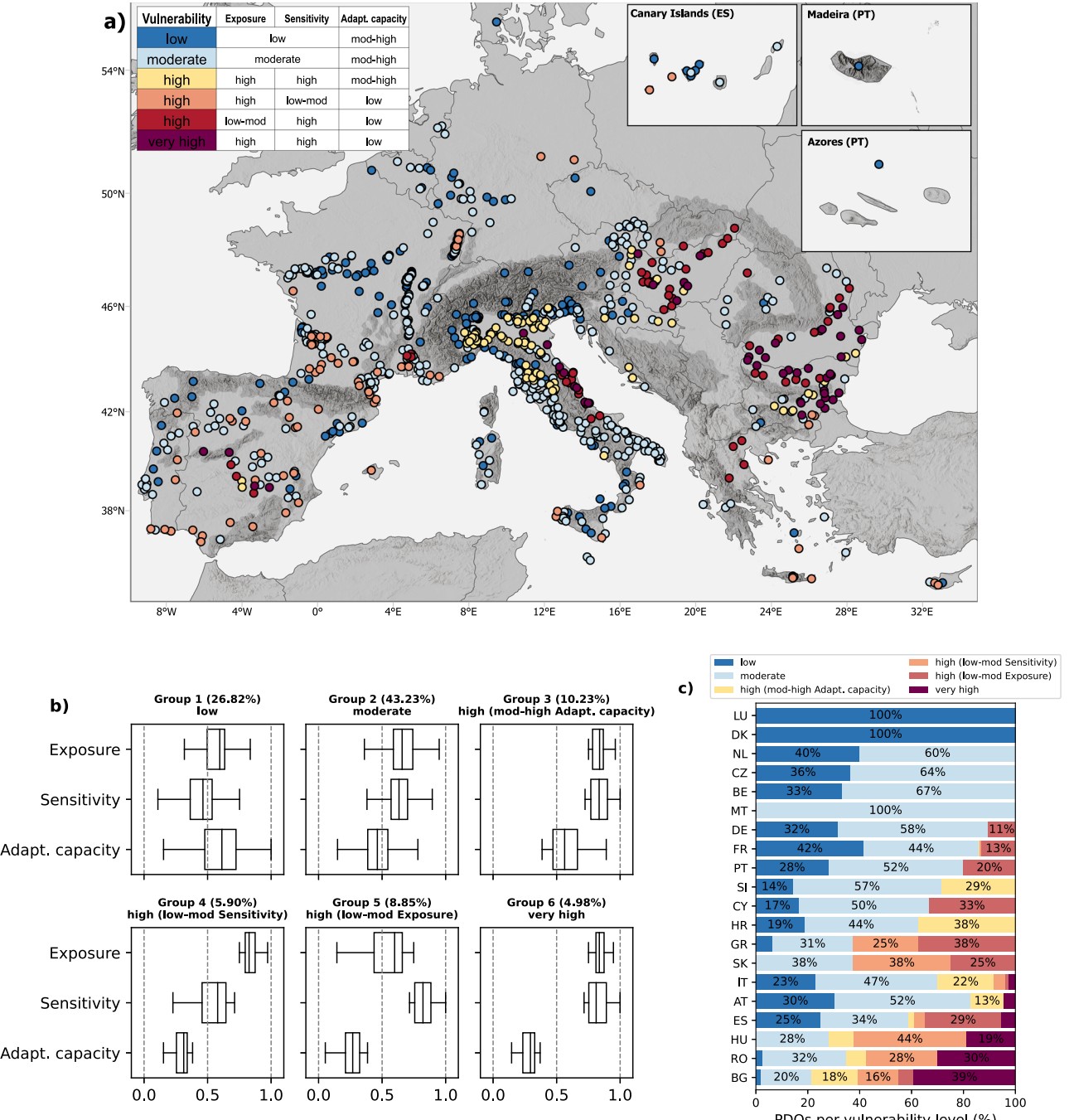

**Fig. 3 | Integrated vulnerability index of European GI regions to climate change. a** The integrated vulnerability index of the wine GIs in Europe. The regions are represented based on their centroids. Dark gray areas refer to mountain regions. Made with Natural Earth. Free vector and raster map data @ naturalearthdata.com. **b** Groups of regions based on the integrated vulnerability index. Each boxplot shows the distribution of exposure, sensitivity and adaptive capacity among the wine regions included in the same group. Total sample size across all groups includes all investigated PDO regions ($n = 1085$). Boxplots show the three-quartile values of the distribution (the line indicates the median and the bounds of the boxes the upper and lower quartiles). The whiskers extend to points that lie within 1.5× interquartile range of the lower and upper quartile. **c** Percentage of wine regions per country based on their integrated vulnerability index. Labels on the bars are only displayed for bars greater than 10%. LU Luxembourg, DK Denmark, NL Netherlands, CZ Czechia, BE Belgium, MT Malta, DE Germany, FR France, PT Portugal, Sl Slovenia, CY Cyprus, HR Croatia, GR Greece, SK Slovakia, IT Italy, AT Austria, ES Spain, HU Hungary, RO Romania, BG Bulgaria.

change the characteristics of the resulting product. Early warning and awareness are therefore critical to successful implementation and to providing the necessary support to prepare eventual amendments in these regions.

While exposure and sensitivity determine the necessity for certain types of adaptations, the adaptive capacity gives insights into the potential for a certain region to implement different strategies.

Strategies such as shifting vineyards to higher elevations or exploiting favorable microclimatic niches, for example, can be very effective in mountain viticultural areas but may not be geographically possible in other regions[53]. Likewise, expanding the possibility for irrigation can be a promising option, but the high economic burden, intensive labor cost, water availability, mechanization requirements, and legal constraints make this option feasible only for regions with sufficient

socioeconomic resources[26]. Regions that have extensive access to resources, including natural, physical, or economic assets, knowledge, and education, as well as the necessary labour force, have the greatest possibility of implementing and elaborating extensive adaptation strategies that would not be feasible in other regions. Especially for regions with strong negative impacts, a high adaptive capacity is therefore critical[57]. However, specific strategies still must be identified and selected on a regional basis, based on detailed assessments. This can be especially important for highly vulnerable regions with a low adaptive capacity, where the timely implementation of adaptation strategies and allocation of resources may become a critical factor.

In addition to the negative climate change impacts in many European wine regions, parts of the European wine sector might also experience benefits from climate change in the near future. This mostly concerns regions with a very low sensitivity, typically located under cool climatic conditions. Strategies in these cases include the identification of already cultivated varieties that could benefit from future climatic conditions and a gradual expansion of their cultivation or introduction of new varieties[58]. However, particular attention should be paid to limit negative ecological effects related to the shift of intensive viticulture into natural and semi-natural systems[14].

## Prospects for further research

European wine PDO regions encompass a remarkable blend of climates, economies, and traditions. As such, they show substantial differences in many aspects of exposure, sensitivity, and adaptive capacity and the related sub-indicators that shape the integrated vulnerability index of each region. The 15 indicators of adaptive capacity, for instance, have been shown to be highly heterogeneous across individual regions leading to nuanced differences, even among close terroirs. Similarly, the interactions between exposure, sensitivity, and adaptive capacity indicators and their contribution to the overall vulnerability of a region are context- and region-driven. At the European scale, such a detailed analysis is currently not feasible, mainly due to critical data gaps related to variety-specific, regulatory and socioeconomic information. This has important implications for future research. Detailed recommendations for improving climate resilience must also consider regional differences in the weight of each indicator and in their combination. This would include an accurate and robust assessment of key variables that are shaping the integrated vulnerability index, such as the response of individual varieties to climate change based on phenological models. Other region-specific characteristics should also be analyzed in greater detail, such as the choice of rootstock type, management of the inter-row vegetation, or the use of a specific training system. Additionally, regional differences in indicator weighting and their contribution to climate change vulnerability must be assessed further to formulate detailed guidelines and policy recommendations based on the uniqueness of each GI area.

Many wine regions as we know them today will change. While the historical concept of terroir emphasizes the specific roles of nature and people in defining wine regulations, adaptation to climate change necessitates flexibility. The rigidity of the GI system, which restricts the exploitation of suitable grape varieties, can contribute to increased vulnerability. Our results present an overview of the vulnerability levels of traditional wine products in Europe and thus provide critical information to build a more climate-resilient GI system. A rigid system for wine production based on a narrow range of grape varieties, fixed management practices, and tight geographical boundaries will likely become obsolete in the next few decades due to the growing impacts of climate change. The GI system must find a way to adapt to climate change and increase its potential for innovation while still preserving the connection between terroir and consumers.

## Methods

### Wine geographical indications

We assessed the vulnerability to climate change of 1085 wine GIs in the European Union. All 1085 regions produce wines labeled as protected designation of origin (PDO). We focused on these regions because the wine products labelled PDO have the strongest link to the production area, i.e., the entire production, processing and preparation process must take place in a specific region[4]. This is in contrast to PGI regions, where only 85% of the grapes have to come exclusively from the PGI area. Additionally, the regulatory scope also covers viticultural and enological practices, including yield regulations, pruning techniques or irrigation, as well as authorized varieties and blend ratios. We focused on PDO regions for which we had sufficient data to calculate all required indicators. The majority of the considered GIs were located either in Italy (35%) or in France (31%), followed by Spain (8%), Bulgaria (4%), Romania (3%), Hungary (3%) and Portugal (3%). The boundaries of the selected regions were taken from Candiago et al.[40,59] and are shown in Supplementary Fig. 1.

### General framework

Grapevines are perennial crops that last for many decades. Successful adaptation strategies, therefore, require long preparation, extensive planning, and careful implementation, as their effects will be apparent for several years. The current capacity for adaptation is a critical factor determining whether a wine region can adapt in time to future impacts of climate change. To combine the current capacity for adaptation with projected climate scenarios, we adapted the vulnerability framework developed by the Intergovernmental Panel on Climate Change (IPCC)[24]. Vulnerability was assessed through exposure, sensitivity, and adaptive capacity (Supplementary Fig. 2). We calculated exposure, sensitivity and adaptive capacity using an index-based approach and developed our indicators based on publicly available statistical and geospatial data. Exposure thereby refers to the general trend and magnitude of climate change in a region and is based on bioclimatic indicators specifically developed for viticulture that describe temperature and precipitation trends under climate change. In contrast, sensitivity refers to how a region is affected by this change in climatic conditions, which depends on specific characteristics of the regional viticultural system with its cultivated varieties. Finally, the adaptive capacity refers to the potential of a region to adapt to changing climatic conditions and includes biophysical as well as socioeconomic aspects. The resulting integrated vulnerability index describes the degree to which a system is susceptible to, or unable to cope with, adverse effects of climate change, also taking into account available resources for adaptation[24]. The vulnerability of a wine region is also directly related to its resilience, as highly vulnerable regions will be less capable of preserving the production of high-quality wine products. Following the guidelines from the original framework, all indicators were standardized onto a relative scale from 0–1, where 0 means the lowest and 1 the highest indicator value of all considered wine regions. The final score is a relative value that compares European wine regions in terms of their individual characteristics and allows the identification of similar regions.

### Exposure

Exposure measures the expected changes in climatic conditions using bioclimatic indicators that are strongly tied to grape berry quality attributes and yields[60], and integrate viticulture-specific information on temperature conditions and water availability during critical stages of vine growth. As such, we assessed the exposure of wine GIs by calculating the change of the Huglin Index[61], Cool Night Index[60] and Dryness Index[62] between the present (1981–2010) and future (2071–2100) reference periods (Supplementary Fig. 3). The present reference period was chosen because it corresponds to the period where most of the PDO regions in Europe were registered

(Supplementary Fig. 4). For the future reference period, we used the ssp370 scenario which corresponds to a temperature increase up to 4 °C until the end of the century[63]. Because we used an index-based approach which is based on relative differences between the considered wine regions, there were no significant differences in the results between the ssp370 and other, more severe, scenarios such as the ssp585. To estimate future climatic conditions, we used an ensemble mean of 5 global climate models ('GFDL-ESM4' 'IPSL-CM6A-LR', 'MPI-ESM1-2-HR', 'MRI-ESM2-0' and 'UKESM1-0-LL'). The models were retrieved from the CHELSA dataset with a 1 km horizontal resolution which was achieved through a statistical downscaling approach[64,65]. Model selection includes the primary models from the Intersectoral Impact Model Intercomparison Project (ISIMIP), which were selected based on their process representation, structural independence, climate sensitivity and performance in the historical period and provide a good representation of the whole CMIP-6 ensemble as they include models with low and high sensitivity[66]. To calculate the exposure for each GI, the change of each bioclimatic index was first calculated for each grid cell. Next, all grid cells within a GI were averaged to obtain a representative value, which was then scaled from 0 to 1 using linear min-max normalization. 0 represents GIs with the smallest changes in climatic conditions and 1 GIs with the greatest changes. Finally, exposure levels were calculated by averaging the scaled values for all three indices for each GI.

## Sensitivity

The sensitivity describes the degree to which a system is affected by climate-related stimuli and is based on the bioregional climate range of the varieties within each GI. The bioregional climate range represents the historic conditions under which each variety has been cultivated and under which it is able to express its best traditional character in the final wine products. It is highly related to how a region is affected by climate change, because the larger the deviation from this range under future conditions, the higher the probability of changes in grape composition and wine style, with negative consequences for the typicity of the PDOs products. We calculated the sensitivity of each GI using the following steps.

Extraction of primary varieties and their cultivation area: we created a database that separates the primary varieties from the additional varieties based on the product specifications of each PDO[67]. Primary varieties are the traditional vine varieties of a region that are primarily used for making the wine products of a GI. In most cases, they were clearly defined in the product specification, however, if there was no specification of primary and additional varieties, we considered all the authorized varieties as primary varieties[68]. We then estimated the cultivation area of the primary varieties for each PDO using the dataset from ref. 69, which contains the cultivation areas for several varieties and regions in Europe. To link this cultivation area to the PDO regions, we first homogenized the variety names using the list of synonyms included in ref. 69, as well as the list of synonyms from ref. 70. This step was carried out at the country level, i.e., for each country, we checked whether all the varieties from ref. 69 were written in the same way as in the PDO regulatory documents. If this was not the case, we first checked whether any synonym from the list of synonyms included in ref. 69 was mentioned in the PDO documents for the corresponding country, otherwise we checked the list of synonyms from ref. 70. In total, we were able to identify the cultivation area for 70% of the varieties listed in the PDO regulatory documents. The remaining 30% of the varieties are either: not currently cultivated within the PDOs, as not all varieties that are authorized are necessarily cultivated; listed under a synonym that does not appear in any of the synonym databases; or were not included in ref. 69.

Finally, we assigned the cultivation area from ref. 69 to the individual PDO regions. In some cases, the cultivation area was already given at the level of PDO regions in which case we linked them directly. In other cases, the cultivation area was given at the level of macro-regions, which may include one or more PDO regions. In this case, we distributed the cultivation area among all the PDO regions in the corresponding microregion, which authorizes a given variety, weighted by the total vineyard area of each PDO, which we derived from the corresponding landcover class from Corine Landcover and OpenStreetMap data.

Definition of the bioregional climate range: we derived the bioregional climate range of each primary variety by linking the varieties to the climatic conditions of the GIs in which they are cultivated, which are based on three bioclimatic indices, weighted by their cultivation area. The bioregional climate range, therefore, represents the historic growing conditions for each variety and not necessarily its growing suitability, as many varieties may also find suitable conditions in regions where they are not currently cultivated. To define the bioregional climate range of each variety, we first classified the GIs into 17 groups based on the Huglin, the Cool Night and the Dryness Index during the period 1981–2010, applying the categorization developed by Fraga et al.[71]. The resulting categorization provides combined information on the bioclimatic characteristics within a region and allows to assign each region to a specific climatic type, ranging from very cool and humid to very warm and dry conditions. Indicators were calculated at 1 km grid resolution using monthly climate data from the CHELSA database[64,65] and then averaged over all grid cells within each region. Next, we calculated a weighted average for each bioclimatic index and variety separately for each region grouping, whereby regions that have a larger cultivation area get a higher weight than regions with a smaller cultivation area. We then identified the bioregional climate range of each variety and bioclimatic index by combining the weighted average with the standard deviation amongst the regions that cultivate this variety within each region grouping. Finally, to validate the resulting climatic ranges, we compared them to climatic ranges for individual varieties reported in other studies. The results showed a very good correspondence to the results reported in previous studies, which shows that our approach is able to accurately approximate historic growing conditions for individual varieties throughout European PDO regions (Supplementary Note 2).

Calculation of the sensitivity: we calculated the sensitivity for each GI based on the difference between current climatic conditions within a GI and the upper limits of the bioregional climate range of its primary varieties. We assumed that once climatic conditions within a GI move outside the bioregional climate range of its primary varieties, the region is increasingly likely to be faced with significant changes in grape composition and wine characteristics[11]. Varieties where current climatic conditions were near the upper limit of their range, therefore, had a higher sensitivity, as a relatively small change in climatic conditions may affect the capacity of the GI to produce traditional wines based on this variety. In contrast, varieties where current climatic conditions were further away from the upper limit of their range had a lower sensitivity. The sensitivity of a GI is given by the average of the sensitivity of all its primary varieties, weighted by their cultivation area. In this way, varieties that are only cultivated over very small areas affect the regional sensitivity less than varieties that have a larger cultivation area. In particular, this enabled us to consider not only species richness in the calculation of regional sensitivity but also to take into account species abundance. Finally, we scaled the sensitivity levels from 0 to 1 using linear min-max normalization, with 0 representing GIs with the lowest sensitivity and 1 representing those with the highest sensitivity.

## Positive effects of climate change on European PDO regions

There are also regions and varieties that can potentially benefit from climate change, e.g., by the increased sugar content of grapes or lower pathogen pressure due to decreased humidity[7,14,27]. To quantify the potential positive effects of climate change in European wine regions, we combined the bioregional climate ranges for the single varieties of

each region with future scenarios for the three bioclimatic indices. We used the present-day weighted average for each variety and region grouping as a reference and calculated if a variety will be shifted closer to this reference value (e.g., experience mostly positive effects) or further away from it (e.g., experience mostly negative effects) under future climatic conditions compared to present-day conditions. We considered the ssp126, the ssp370 and the ssp585 scenarios for the periods 2041–2070 and 2071–2100 to compare potential benefits from climate change in European PDO regions under different scenarios.

### Adaptive capacity

To assess the adaptive capacity of the GIs, we collected several indicators related to their individual socioeconomic and biophysical characteristics. We identified 15 indicators related to five dimensions of adaptive capacity in the context of viticulture. For the full list of considered indicators, their source and calculation method, please refer to Supplementary Note 1. To calculate the adaptive capacity of each wine GI, all the considered indicators were first scaled to a range between 0 and 1 using the 5th and 95th percentiles as lower and upper thresholds to reduce the influence of outliers[72].

To analyze potential differences in the importance of the adaptive capacity indicators, we carried out a survey including five PDO regions across Europe. The regions were selected to cover a wide range of the topoclimatic conditions and wine styles that can be found in the winegrowing regions of Europe. As such, they include the Mediterranean region of Maremma Toscana in Italy, the mountainous regions of Douro in Portugal as well as Südsteiermark in Austria, the cool-climate region of Moselle Luxembourgeoise, and the continental region of Târnave in Romania (Supplementary Fig. 13a). To analyze the importance of the adaptive capacity indicators in these regions, we contacted an expert for each region and asked them to rank the indicators using an analytical hierarchy process (AHP)[73,74]. This approach has already been used in previous studies in the context of climate change adaptation[75–77] and consists of pairwise comparisons of indicators where the experts assess their importance for climate change adaptation in the respective region. After the initial assessments, all the experts were asked to adjust their statements until the final consistency ratio dropped below 10%. The results from the survey clearly show that the ranking of the indicators is highly different amongst our analyzed regions (Supplementary Fig. 13c–g). Consequently, there is a very weak correlation between the indicator weights of the individual regions (Supplementary Fig. 13b). These results indicate that indicator importance strongly depends on the characteristics of individual regions. Using a single set of weights across all European PDO regions might, therefore, be problematic and introduce significant biases for many of them. Therefore, we decided to use an equal weight during our main analysis because this constitutes the most neutral approach.

The adaptive capacity was therefore calculated by averaging the scores of the indicators and then scaled again to a range between 0 and 1 using linear min-max normalization, with 0 representing regions with the lowest adaptive capacity and 1 representing those with the highest adaptive capacity:

$$AC_i = \frac{X - Q_5}{Q_{95} - Q_5} \qquad (1)$$

$$AC = \frac{1}{n}\sum_{i-1}^{n} AC_i \qquad (2)$$

where $AC_i$ represents the adaptive capacity score of indicator $i$, $X$ is the indicator value for a particular GI, $Q_5$ and $Q_{95}$ are the 5th and 95th quantiles, respectively, and $AC$ is the final adaptive capacity indicator.

### Integrated vulnerability index

We analyzed the integrated vulnerability index of each GI by comparing the three indicators of exposure, sensitivity, and adaptive capacity. This approach allowed us to group the winegrowing regions into six different groups, each of them consisting of regions with comparable characteristics. We first classified each of the three indicators as low, moderate, or high using percentiles (0–33%, 33–66% and 66–100%) and then assigned each region to a vulnerability level based on the following rules:

Very high vulnerability: regions classified as having high exposure and sensitivity and low adaptive capacity

High vulnerability: regions where two indicators were classified as high (or low in case of adaptive capacity)

Low vulnerability: regions where at least two indicators were classified as low (or high in case of adaptive capacity)

Moderate vulnerability: remaining regions not assigned to another vulnerability level.

### Reporting summary

Further information on research design is available in the Nature Portfolio Reporting Summary linked to this article.

## Data availability

The dataset containing the primary and additional varieties cultivated in each PDO generated in this study has been deposited in the Zenodo database under accession code https://doi.org/10.5281/zenodo.7257126. The dataset containing the exposure, sensitivity and adaptive capacity indicators generated in this study has also been deposited in the Zenodo database under accession code https://zenodo.org/records/10410972. The sources for the data underlying the individual indicators are provided in the supplementary materials file alongside the description of each indicator (Supplementary Note 1).

## Code availability

Data processing was conducted using Microsoft Excel (version 2405), QGIS 3.34 and ArcGIS Pro 3.2.2. No custom code was generated.

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

## Acknowledgements

S.T. PhD grant was co-financed by the 'Südtiroler Sparkasse Foundation' with additional funding received from the 'Fusion Grant'. S.C. PhD grant was co-financed by the European Regional Development Fund through the Interreg Alpine Space Programme ('AlpES | Alpine Ecosystem Services – mapping, maintenance, management', project number ASP 183), and the Interreg V-A ITA-AUT programme (REBECKA, project number ITAT1002). H.F. thanks the Portuguese Foundation for Science and Technology (FCT), for UIDB/04033/2020 (https://doi.org/10.54499/UIDB/04033/2020), LA/P/0126/2020 (https://doi.org/10.54499/LA/P/0126/2020) and 2022.02317.CEECIND (https://doi.org/10.54499/2022.02317.CEECIND/CP1749/CT0002). The authors thank Fiona Nevzati for her help with the graphical presentation of the figures included in the manuscript.

## Author contributions

S.T. and S.C. contributed equally to this work. S.T., S.C. and L.E.V. conceived the study and designed it with, C.G. and H.F. jointly. S.T. and S.C. developed the exposure and sensitivity indicators and S.T., S.C. and T.M. together developed the adaptive capacity indicators. S.T. performed the analysis. C.G. and L.E.V. supervised the project. S.T., S.C. and L.E.V. wrote the manuscript with the contribution of all co-authors.

## Competing interests

The authors declare no competing interests.
