## [Peer Review File · Nature Communications]

REVIEWER COMMENTS

Reviewer #1 (Remarks to the Author):

In ms. NCOMMS-22-45849 entitled "Climate resilience of European wine regions", the authors analyze the extent to which climate change will affect European viticultural regions, and how well-equipped these regions are to withstand and adapt to projected changes. The authors group and rank regions according to their differential vulnerabilities and propose adaptations pathways tailored to each grouping. The paper is overall well written and focuses on a hot topic that has attracted much research attention lately. Further, some of the results could aid guiding future decisions by policy makers to successfully adapt a critical economic sector to unwanted effects of climate change. While I was initially excited to review a much-needed paper that quantifies climate change impacts in European wine growing regions, I was underwhelmed after reading it due to critical assumptions made, methodological issues and lack of clarity and justification for critical methodological decisions. Despite its potential, the ms. has several flaws that should be addressed by authors to meet Nature communication's standards. I outline a couple major concerns and several minor ones below.

Major concerns

1. A major conceptual problem has to do with the critical assumption that any change in climate conditions within viticultural GI's is or will be negative. This assumption is problematic because clearly, some climate change driven shifts in climatological conditions may be particularly beneficial for some agricultural regions (e.g., decreased humidity leading to less fungal infections in susceptible areas, or warmer temperatures leading to higher alcoholic contents in regions where given varieties do not accumulate enough sugar frequently). Most literature conducting agricultural forecasts accounts for both negative and positive outcomes from climate change (e.g., Rosenzweig et al. 2013:PNAS 111 (9) 3268-3273). Not acknowledging so lacks rigor as it hides part of the full picture and, also, may contribute to a biased perception that outcomes of climate change can solely be negative or catastrophic. I understand the point of assuming that whatever climate a given region has experienced historically, is what got that region protected and, any changes to it would be bad. Even under such assumption, the choice of the 1980-2010 period as a historical reference, may not necessarily reflect the historical climate conditions that led GI's to become what they are since it already includes some effects of climate change. Ideally, I would suggest authors to also account for any potential benefits of climate change in their analyses. I realize doing so may be rather complicated as it would involve major re-analyses. So, at least some in depth discussion of the assumption and some sensitivity analyses (e.g. how do results change when not all shifts in climate are considered detrimental?), would be needed.

2. A second conceptual-methodological issue has to do with the so-called biogeographical niche. First, the terminology used does not seem appropriate, less so bearing in mind the strong connotations and implications that the terms "biogeographical" and "niche" have in the ecological and climate change forecasting literature. These terms are most frequently referred to jointly in the Species Distribution Modelling or Ecological Niche Modelling literature and it would be easy to have readers misled and confused expecting one of those models. I wonder if coining a new term is needed. Since what their index actually captures is the range of climate conditions experienced by a given primary variety across the GI's (within a group) where it is grown, I would encourage authors using more informative terminology, e.g., bioclimatic extent, bioclimatic capacity, bioclimatic envelope, bioregion-constrained climate range, etc. Second, and more important, the calculation of the index may fail to precisely capture the climate conditions experienced by primary varieties where they are planted because climate indices used are computed along fixed temporal windows, thus dismissing varietal physiology and phenology. The latter are what determines biologically effective temporal windows defining actual climate experienced by different varieties in different regions and not accounting for it can lead to important biases (discussed elsewhere, see e.g., Van Leeuwen, et al., PNAS 110(33), E3051-E3052). For example, following the "biogeographic niche" approach, two varieties, one very early and one very

late grown in the same two regions within a region grouping, would be assigned the exact same niche even though they may experience rather different climate conditions at relevant developmental stages. The authors are aware of this issue and touch briefly on it but fail to soundly justify their choice. Again, there is not a straightforward way around this issue in absence of phenology data for each variety at each location. Yet, at least for one or few varieties for which there is good information, authors could run a sensitivity test that tells whether their approach underestimates region sensitivity. Doing so would not necessarily involve having information of a variety across all sites where it is grown, but perhaps focusing on the regions-years known to be of the highest quality for a given variety.

3. The rationale underlying the index for adaptive capacity (L133) is not fully clear and seems somewhat arbitrary. Also, how are temperature and water included as indices? Why would temperature variability within a region provide it with more adaptive ability? What are the units? In general, more details would be needed in the main text to allow following how results were generated and why critical methodological decisions were made. An additional issue with the adaptive capacity index is that all variables are weighted the same and it seems plausible that some may not be problematic at all in some regions. While it would be unfeasible to have data allowing to weight the variables at each GI, it would be very interesting to send a small survey to a few regions asking to growers which, in their opinion, are the most critical dimensions constraining future adaptation, and compare with their results. This is just an idea, overall, I found the effort to integrate the human-social-economic dimension highly valuable.

Minor comments

I14. yield

I34. I'd nod to the fact that Terroir usually also refers to climate

I56. Cite also ref. 21

I67. what is the difference between natural and physical?

I72. what do you mean by biogeographical niche? The niche concept has broad implications in ecology and elsewhere and thus should be used with caution.

I80. beware of overplaying the reach and potential impact of your results.

I88. mountainous?

I117. Perhaps not only increasing diversity "uncritically" but selecting which cultivars harbor more future interest for a specific place and using those for as an informed as possible varietal turnover
I126-127. then it would be valuable to show the standard deviations or some measurement of error around those means x-values.

I154. please state what are those benefits

Fig 4b is rather confusing and hinders relevant information on the uncertainty or error around adaptive capacity estimates relative to each dimension

I264. Why the choice of reference periods? It would be useful to know what degree of warming these periods represent (with respect to pre-industrial temperatures) to make them comparable with previous studies.

I268. How do you justify the choice of these specific set of models? Please make whatever the justification is, explicit.

I275. How is model uncertainty dealt with? Since as authors recognise uncertainty is inherent to our inability to exactly predict future climate trajectories, such uncertainty should be explicitly incorporated and reported along with the results. For example, authors could add, even if only as appendices, maps of uncertainty showing where model predictions are more uncertainty (e.g., an uncertainty map for exposure, etc.).

I278. Why any change in climate would be negative (please see major comment related to this)? More so, using a reference period for present time that does not reflect historical conditions driving a given GI to become important (i.e., 1980-83 is often considered as the period when climate warming started to manifest more clearly).

I286. The 'biogeographic niche' may be problematic (see major comment related to it), more so

when referring to a well-delineated geographic region (a GI). Why not using bioclimatic niche or bioclimatic space, instead?

l298. More details on the specific categorization scheme should be provided here without having to go back to Fraga et al. paper.

Reviewer #2 (Remarks to the Author):

I really like the idea of this study – it is certainly pertinent to consider the vulnerability of European wine regions and would be of great interest to readers of Nature Communications. However, I have some significant concerns about the ways in which the various scores have been combined.

Firstly, in relation to how exposure and sensitivity were calculated. Exposure is estimated by examining how mean annual temperature and rainfall are expected to change under climate change. However, there is no real justification provided for the selection of these variables, and the default assumption would actually be that cultivars are more sensitive to a range of other bioclimate metrics. Indeed, and confusingly, other more relevant metrics are used to define the biogeographic niche of the species and to assess sensitivity. The blend of different variables is confusing, poorly explained, and unwarranted. A far more logical approach would have been to use a niche-based modelling approach (aka climate envelope model / species distribution model). In this way, the likely impact of climate change could have been properly quantified as opposed to relying on some rather arbitrary combinations of sensitivity and exposure scores.

Second, in the calculation of adaptive capacity, 15 indicators are calculated, and after rescaling to give them a consistent range, they are then just given equal weighting to calculate a total adaptive capacity. Again, however, this seems rather arbitrary, as one would surely expect some to be of greater significance than others. A better approach would have been to consider more carefully how each could or should be weighted and combine the scores accordingly.

Similarly, the calculation of total vulnerability as exposure + sensitivity – adaptive capacity is completely arbitrary. There is nothing in the methods to suggest that these have a common scale that make combining them in this way justifiable. While I can see the attraction of having a simple overall metric, it is simply misleading to present it in this way, and it would be far more appropriate to retain these as separate axes.

In short, this paper has some very interesting and noteworthy semi-quantitative findings, that really do not lend themselves to being presented as wholly quantitative in the way the authors have done.

Reviewer 3

Overview and significance of work:

The authors have used an extensive dataset of biophysical, social and economic factors to assess the vulnerability of current wine growing regions to the effects of climate change. The regions are defined by PDOs (Protected Designation of Origin) in some European countries where the assumption is current viticulture is suited to current climate. Of significance is the area covered, and the combination of the biophysical, social and economic factors to assess vulnerability integrating key components from the latest IPCC report. This has not been previously reported for grapevines (often work has focused on the biophysical suitability), representing a novel and comprehensive multidisciplinary approach. The clustering method used enable the authors to find an integrative way to bring together these factors, and resulted in identifying areas of low, medium and high vulnerability by the metrics they used.

The significance of this work is it is the first objective assessment of vulnerability carried out on a scale of significance (several European countries) for viticulture production in the context of climate change. The conclusion that if PDOs are and remain rigid, then resilience to climate change will be reduced and these systems need to be more flexible is relevant. If the key methodological points were addressed, then this would represent a significant advancement in characterising vulnerability. However, the work does not offer solutions (pathways) to addressing the vulnerability. The advance in the field will come from clearly linking the vulnerability outcomes to pathways in an objective manner (see comments below).

The text was clear and accessible and previous work adequately cited. In terms of references and comparing the work to established literature, information on vulnerability assessment for other crops or species is missing and should be considered.

References were relevant and supported the points raised. It may be that some of the methods considerations outlined below could be addressed also by inclusion of relevant references.

Interpretations and validity:

The data interpretations with regard to low, medium and high vulnerability is clear (aside from some important concerns about the methods outlined below which may influence the outcomes of their vulnerability assessment). However, the interpretation for pathways forward on page 7 currently is not supported by the results. For example, nothing in the Figure 3 or its analysis tells us directly that high vulnerability regions need the pathway of biophysical strategies of vineyard management and grape diversification and relocation to mitigate this vulnerability. This interpretation is not directly possible from the analysis presented so far. To draw such interpretation the authors would need to objectively evidence the relative importance of the biophysical components in determining the vulnerability of high vulnerability areas. The authors would need to develop a metric/ or statistical framework for this to be supported. The same comments could be made for the other two pathways outlined in page 7. Furthermore, if high vulnerability areas are predominantly dependent on these biophysical factors, what is the need to include the adaptive capacity in assessing vulnerability for this situation? In addition, the suggestion to shift production in space is ok but the manuscript failed to address that higher elevation also mean aspect may differ which could also have a large impact on wine style for retaining current PDO characteristics.

Data and methodology:

The framework of the methodology is sound and very clearly explained. However, some method aspects should be reviewed.

The details for the various components may not be the best way of assessment the impact. Exposure is also equally weighted to adaptive capacity in the vulnerability assessment but substantially less detailed. Notably the biophysical metrics of annual mean temperature and precipitation over-simplify the situation for calculating "exposure" relative to what is normally used in viticulture and crop production, a calculation of Growing degree-days and specific precipitation patterns at key times of the year for grape production. It is unclear why these metrics were chosen given the depth of work that already exists using more relevant bioclimatic indices or phenological modelling approaches. A wine base study should be at minimum, operating using Growing degree-days (temperatures relevant to the growing season of grapes) and precipitation during this period. Without doing the analysis, it is unclear whether this would change the vulnerability classifications. Why were heat stress and drought metrics also not considered?

This also raises the following question: Why the Huglin index and not phenological models were employed for sensitivity analysis where there is quite big differences already described among cultivars by phenological models.

While the rationale and results of clustering is justified, the weighting of exposure, sensitivity and adaptive capacity in the vulnerability index need to be justified. How and why did each component get equal weighting especially when adaptive capacity amalgamates many metrics and the other two only a few?

The origin of the mechanisation index is unclear. One could argue a high dependency on machinery increases adaptive capacity as it leads to increased mechanisation and reduced labour costs

(particularly in settings that may score low for Ageing index and dependency ratio). Description does not clearly reflect the calculation.

Analytical approach:

The strengths of the approach is integrating the three metrics in the vulnerability assessment and clustering results. Note that the supplementary material, link 74- 75 state "87% and 75% of the regions obtain the same scoring as when using 7 or 8 groups, respectively", which means for example for 75%, that 1/4 of regions get a different score depending on the number of groups in the clusters. This represents substantial differences in the groupings of vulnerability depending on the number of clusters used. Given these differences the methods to date do not present an adequate justification of selecting six groups so further justification of this is required.

There was also an assumption that any change resulted in negative effect because these changes alter the specific biophysical conditions that are needed to produce a certain GI wine. However, between season variability leads to interannual differences in production. So negativity should be assessed relative to this.

Suggested improvements for revision:

Most key suggestions are covered under the feedback for 'data and methodology' and 'analytical approach'. To summarize:

- the metrics used for exposure and sensitivity should be reassessed and may need testing of alternates (outlined in method section). In particular, yearly average temperature is too broad given the current knowledge of bioclimatic indices and phenological models.

-The decision to include six clusters needs better justification as highlighted under the section 5.

Analytical approach

-The weighting of the difference components in the vulnerability index needs justifying

-The link between the vulnerability results and proposed pathways needs to be supported by further analysis.

Further minor suggestions for improvement are:

- The term Europe in the heading is too broad. For example, key countries of Germany, Austria, Switzerland, and Greece are missing. The specifics rather than using the term 'Europe' also needs to be addressed in the introduction.
- Line 42 'substitute grape candidate' – change to grape to cultivar
- Line 45- 'resilience' this should be defined in terms of what the authors consider resilience relates to
- Line 135. It is interesting that the adaptive capacity high where exposure was high. What is the explanation?
- Figure eight suggested vulnerability is predominantly climate driven, how does using VI enable us to demonstrate this importance of integrating adaptive capacity?
- More detail is required for lines 302- 3-5 in the methods. "Linking all the appellations within a group that grew a specific variety to that variety". How was this combined with the biogeographical niche of a GI?
- Lines 307- How was the upper limit defined?
- Education level= why *2 for educated? Justification for the value of this multiplier?
- Line 227- should be explicit these are in non-grape related works (i.e. the approach is a more universal thing)
- Figure 2-colours in b and c should match
- Abbreviations in figure 2 should be explained in the figure caption
- Significant areas on Sup Fig 1 not included
- Table 1 repeats Figure 2. Delete table
- Suppl. Figure 2-1 hard to read
- Suppl. Table 2-1 not clear

Reviewer #4 (Remarks to the Author):

This is a very welcome and well-written piece dealing with a vulnerability assessment of European wine regions, under the EU GI system, in relation to climate change to enhance the understanding of which regions are threatened the most by climate change and support the development of potential adaptation pathways to strengthen their resilience.

The authors make this assessment based on the registered wine GIs (PDO/PGI) system and respective regions. The authors specifically state: "Vulnerability is driven by the rigidity of the GI system, which restricts the exploitation of suitable bioclimatic conditions and existing grape cultivar diversity, as well as contextual deficiencies, such as limited socioeconomic resources".

To assess the climate change vulnerability, the authors adapted the framework developed by the Intergovernmental Panel on Climate Change (IPCC) in which vulnerability is assessed as a function of exposure, sensitivity, and adaptive capacity. To operationalize the assessment the authors defined: (i) exposure as the expected changes in climatic conditions, including temperature and precipitation, (ii) sensitivity as the degree to which a system is affected by climate related stimuli, based on the biogeographical niche of each wine region, and (iii) adaptive capacity as how well a wine region can adapt to changing climate conditions, considering five distinct dimensions (financial, natural, physical, social, and human).

It is worth noting that the GI system (PDO/PGI) allows for amendments to the product specifications of the protected GIs for wine as specified in the GI legislation the authors add (reference 21). So, the EU GI framework allows for amendments. In fact, the current GI legislation (i.e., food products) is also being amended to cope with sustainability issues, is there also any ongoing modification to the EU (PDO/PGI) regulation for wines? Authors such as Cupri 2020, Edelmann et al. 2020, Marescotti et al. 2020, Quiñones-Ruiz et al. 2018, Teil 2020 (for wine), have analyzed adaptations for safeguarded GIs, mostly involving food products (i.e., fruit and vegetables, cheese), in which producers have to justify why they modify the "rules of the game", namely the product specifications. Some of the amendments are also related to climate change. It would be worth to specifically show (approved) amendments for wine GIs in general and specifically those dealing i.e., with changing climate conditions, increasing/decreasing geographical area, ... to also observe how producers are formally justifying and coping with external (or internal) factors to their production systems and discuss them with the very interesting findings the authors provide at regional level (1174 wine regions? Or wine GIs?).

The authors state: "we assessed the climate change vulnerability of 1174 wine regions in Europe by explicitly considering their biophysical and socioeconomic characteristics and their regulatory specifications". "Our results provide the first overview of the vulnerability levels of traditional wine products in Europe combined with adaptation pathways that provide critical information to build a more climate-resilient GI system." Just for clarification, do you specifically talk about 1174 wine regions or wine GIs based in European regions? how many GI wines are you covering in case you deal with 1174 wine regions? in case this data is available and differentiation possible. Does the analysis make a differentiation between PDO and PGI wines or include a differentiation concerning the adaptation pathway and certain trend based on the GI schemes (PDO/PGI)? Do the adaptation strategies showed in the very interesting grouping of pathways specifically deal with already approved amendments just to be precise with the accepted modifications to the production rules and the wording used related to the EU regulation?

This is just to further understand the rigidity the authors aim to show.

Perhaps for sake of clarity, please stick to the GI system, since this is the system the authors deal with, and avoid the term appellation (which is usually used for the national (GI) French system, unless you talk about French appellations) and perhaps briefly state the difference between PDO and PGI to show the most rigid scheme.

It was not possible to open the links below, please kindly update them if likely:

<https://commission.europa.eu/food-farming-408fisheries/food-safety-and-quality/certification/quality->

[labels/geographical-indications-409register/details/EUGI00000000861](https://commission.europa.eu/food-farming-409fisheries/food-safety-and-quality/certification/quality-labels/geographical-indications-409register/details/EUGI00000000861)

<https://commission.europa.eu/food-farming-411fisheries/food-safety-and-quality/certification/quality-labels/geographical-indications-412register/details/EUGI00000002663>

<https://commission.europa.eu/food-farming-481fisheries/food-safety-and-quality/certification/quality-labels/geographical-indications-482register/details/EUGI00000001605>

https://agriculture.ec.europa.eu/farming/geographical-501indications-and-quality-schemes/geographical-indications-food-and-drink/chianti-pdo_en

Minor: the paper is very well-written, just double check if only British English is used (i.e., analyzing).

Finally, congratulations for this piece!

REVIEWER COMMENTS

Reviewer #1 (Remarks to the Author):

In ms. NCOMMS-22-45849 entitled “Climate resilience of European wine regions”, the authors analyze the extent to which climate change will affect European viticultural regions, and how well-equipped these regions are to withstand and adapt to projected changes. The authors group and rank regions according to their differential vulnerabilities and propose adaptations pathways tailored to each grouping. The paper is overall well written and focuses on a hot topic that has attracted much research attention lately. Further, some of the results could aid guiding future decisions by policy makers to successfully adapt a critical economic sector to unwanted effects of climate change. While I was initially excited to review a much-needed paper that quantifies climate change impacts in European wine growing regions, I was underwhelmed after reading it due to critical assumptions made, methodological issues and lack of clarity and justification for critical methodological decisions. Despite its potential, the ms. has several flaws that should be addressed by authors to meet Nature communication’s standards. I outline a couple major concerns and several minor ones below.

Major concerns

1.1 A major conceptual problem has to do with the critical assumption that any change in climate conditions within viticultural GI’s is or will be negative. This assumption is problematic because clearly, some climate change driven shifts in climatological conditions may be particularly beneficial for some agricultural regions (e.g., decreased humidity leading to less fungal infections in susceptible areas, or warmer temperatures leading to higher alcoholic contents in regions where given varieties do not accumulate enough sugar frequently). Most literature conducting agricultural forecasts accounts for both negative and positive outcomes from climate change (e.g., Rosenzweig et al. 2013:PNAS 111 (9) 3268-3273). Not acknowledging so lacks rigor as it hides part of the full picture and, also, may contribute to a biased perception that outcomes of climate change can solely be negative or catastrophic. I understand the point of assuming that whatever climate a given region has experienced historically, is what got that region protected and, any changes to it would be bad. Even under such assumption, the choice of the 1980-2010 period as a historical reference, may not necessarily reflect the historical climate conditions that led GI’s to become what they are since it already includes some effects of climate change. Ideally, I would suggest authors to also account for any potential benefits of climate change in their analyses. I realize doing so may be rather complicated as it would involve major re-analyses. So, at least some in depth discussion of the assumption and some sensitivity analyses (e.g. how do results change when not all shifts in climate are considered detrimental?), would be needed.

We thank the reviewer for raising this valuable point. We agree that climate change, to some extent, can also have positive effects for certain winegrowing regions and varieties. In fact, our sensitivity analysis takes account for these positive effects, although so far indirectly. Low sensitivity levels, for example, indicate regions that have a higher number of varieties that can potentially benefit from climate warming, as they are cultivated under comparatively cool/humid climatic conditions. However, so far, our approach did not account for the distribution of varieties across different PDOs within a region grouping, which would allow us to more accurately derive variety-specific climate ranges. To improve this aspect and explicitly show potential positive effects of climate change on varieties and regions, we i) refined the benchmark of the climatic ranges for the single varieties by including explicit information about their cultivation area in the different PDO regions (for more details, please refer to comment 1.2) and ii) assessed whether the different varieties are moving towards, or are moving away from these reference benchmarks under different warming scenarios.

We therefore have extensively re-visited the manuscript, aimed at discussing in a clearer way also the potential positive effects of climate change on European PDO regions under different future scenarios and how they are related to the climate change sensitivity and vulnerability of the PDO regions. In particular, we showed which varieties (and regions) will likely benefit from future warmer growing conditions compared to their current conditions (Figure 1d and Supplementary Figure 6). We also incorporated these findings into the result section and the discussion on potential future pathways (Lines 111-115 and lines 220-227 and lines 270-276). Accordingly, both the method section and the supplementary material have been revised and updated to include the new methodological approach (Lines 400-410).

1.2 A second conceptual-methodological issue has to do with the so-called biogeographical niche. First, the terminology used does not seem appropriate, less so bearing in mind the strong connotations and implications that the terms “biogeographical” and “niche” have in the ecological and climate change forecasting literature. These terms are most frequently referred to jointly in the Species Distribution Modelling or Ecological Niche Modelling literature and it would be easy to have readers misled and confused expecting one of those models. I wonder if coining a new term is needed. Since what their index actually captures is the range of climate conditions experienced by a given primary variety across the GI’s (within a group) where it is grown, I would encourage authors using more informative terminology, e.g., bioclimatic extent, bioclimatic capacity, bioclimatic envelope, bioregion-constrained climate range, etc.

We understand the concern raised by the reviewer about the use of the term “biogeographical niche”. Our initial selection was driven by the intention to clearly distinguishing our approach from the bioclimatic indices that are commonly used in the field (i.e., general classifications purely based on thermal conditions), but we agree that this might be misleading. Therefore, to avoid any ambiguity, we have now changed the term “biogeographical niche” to “bioregional climate range”, which in our opinion also better conveys the innovative regional approach we used to derive the representative climatic ranges across multiple regions.

1.3 Second, and more important, the calculation of the index may fail to precisely capture the climate conditions experienced by primary varieties where they are planted because climate indices used are computed along fixed temporal windows, thus dismissing varietal physiology and phenology. The latter are what determines biologically effective temporal windows defining actual climate experienced by different varieties in different regions and not accounting for it can lead to important biases (discussed elsewhere, see e.g., Van Leeuwen, et al., PNAS 110(33), E3051-E3052). For example, following the “biogeographic niche” approach, two varieties, one very early and one very late grown in the same two regions within a region grouping, would be assigned the exact same niche even though they may experience rather different climate conditions at relevant developmental stages. The authors are aware of this issue and touch briefly on it but fail to soundly justify their choice. Again, there is not a straightforward way around this issue in absence of phenology data for each variety at each location. Yet, at least for one or few varieties for which there is good information, authors could run a sensitivity test that tells whether their approach underestimates region sensitivity. Doing so would not necessarily involve having information of a variety across all sites where it is grown, but perhaps focusing on the regions-years known to be of the highest quality for a given variety.

This is an important point, and we agree with the reviewer that differences in phenology are critical for the impacts of climate change impacts on vine varieties. Unfortunately, data on the phenological development of vine varieties are extremely limited; we do not have information on the phenological characteristics for the vast majority of the over 1000 varieties currently registered in European PDO

regions. Likewise, fine scale spatial data for the precise location of these varieties within the PDOs is largely lacking, which would be essential for a more detailed phenological/climatic characterization.

However, to better account for the concerns of the reviewer, we have extensively revised the climate change sensitivity assessment and now propose an extended approach that combines the PDO spatial data and regional statistical data on the cultivated area per region and variety, taken from ref¹. This allowed us to weight the importance of individual varieties within region groupings and, in our opinion, substantially improves the climate ranges assigned to each variety. Moreover, we also carried out a validation of this approach by comparing our varietal climate ranges with those identified by previous studies in the Supplementary Methods.

The new methodological approach in detail:

Using the statistical dataset on cultivation area per region and variety, the characteristic range of climatic conditions is now calculated separately for each variety in a PDO region based on its actual cultivation area. Thereby we get a better differentiation between early and late-ripening varieties, e.g., two varieties cultivated in the exact same regions now get different ranges if their cultivation area in these regions is different. As early ripening varieties tend to have larger cultivation areas in cooler regions, their range is shifted to cooler climatic conditions, whereas later-ripening varieties are mostly cultivated in warmer regions, and thus their range is shifted towards warmer climatic conditions. The inclusion of the cultivation area for each PDO region and variety thus provides a substantial improvement that allows us not only to differentiate better between single varieties but also to consider their relative abundance within the PDO regions when determining the sensitivity to climate change. For example, the sensitivity of a region where most of the cultivation area is occupied by a single variety now is mostly influenced by the sensitivity of this specific variety, while in our previous approach, all varieties had an equal influence. Although we acknowledge that our approach could be further improved with more accurate phenological data and a more realistic representation of the local context if performed at a regional or even local scale, we believe that this approach represents an acceptable compromise, especially given the scope of our study and the high demand for pan-European estimates.

In line with these changes, we modified several parts of our manuscript: i) we updated the Material and Methods section of the main manuscript to describe our new approach to derive the sensitivity to climate change (Lines 353-399); ii) we expanded the supplementary material by including a validation of our modified approach which compares our climatic ranges to previously published climatic ranges for several national and international varieties (supplementary material lines 80-106); and iii) we added a new section in the supplementary material, describing how we derived the cultivated area per variety for the PDO regions from the newly introduced statistical dataset (Lines 61-79). Furthermore, we acknowledged the data limitations that prevented us from including detailed phenological information in our analysis; we now extended this description in the discussion of climate change sensitivity to better justify the choice of our approach (Lines 121-129).

1.4 The rationale underlying the index for adaptive capacity (L133) is not fully clear and seems somewhat arbitrary. Also, how are temperature and water included as indices? Why would temperature variability within a region provide it with more adaptive ability? What are the units? In general, more details would be needed in the main text to allow following how results were generated and why critical methodological decisions were made. An additional issue with the adaptive capacity index is that all variables are weighted the same and it seems plausible that some may not be problematic at all in some regions. While it would be unfeasible to have data allowing to weight the variables at each GI, it would be very interesting to send a small survey to a few regions asking to growers which, in their opinion, are the most critical

dimensions constraining future adaptation, and compare with their results. This is just an idea, overall, I found the effort to integrate the human-social-economic dimension highly valuable.

Thank you for pointing this out. Following your recommendation, we expanded the explanations on the adaptive capacity index (Lines 143-155) and extended Table 1 in the main manuscript to provide more detailed information on the individual indicators, including their units. However, we decided to retain the detailed indicator description (and the specific rationale for their selection) in the supplementary material because it would be too extensive for the main document. To facilitate understanding, we now explicitly direct the reader to the methods section and the supplementary material where the rationale behind each indicator is further described. We also renamed some of the indices to make it clearer to which processes they are related and why they were included in the calculation of adaptive capacity.

Regarding the weights of the indicators, we agree that some indicators might be more relevant for some regions. To show how indicator importance changes in different regions, we have conducted a survey among a panel of experts across five different wine-growing regions in Europe (see also comment 2.2). The survey revealed significant differences in indicator importance and shows that their importance strongly depends on individual characteristics of the wine regions. At the continental scale, indicator weighting is therefore problematic, as the characteristics of single regions differ significantly, and we therefore used equal weights in our main analysis. We now openly discuss our reasoning for the selection and weighting of the indicators in the main text (lines 418-422) and included the detailed results from our survey in the supplementary material (lines 116-137).

Minor comments

1.5 l14. Yield

Corrected

1.6 l34. I'd nod to the fact that Terroir usually also refers to climate

Corrected

1.7 l56. Cite also ref. 21

We assume the reviewer refers to ref. 12 (Hannah et al. 2013) and updated the citation by inserting this article.

1.8 l67. what is the difference between natural and physical?

Thank you for this point. Physical refers to the availability of physical assets (such as infrastructure) in a region, while natural refers to topoclimatic characteristics of a region. To avoid confusion between the two terms, we replaced natural and physical with 'bio-physical'.

1.9 l72. what do you mean by biogeographical niche? The niche concept has broad implications in ecology and elsewhere and thus should be used with caution.

We replaced the term "biogeographic niche" with "bioregional climate range" throughout the manuscript (see also answer to comment 1.2).

1.10 l80. beware of overplaying the reach and potential impact of your results.

We removed this sentence in the new version of the manuscript.

1.11 l88. mountainous?

Corrected

1.12 I117. Perhaps not only increasing diversity "uncritically" but selecting which cultivars harbor more future interest for a specific place and using those for as an informed as possible varietal turnover

We rephrased this sentence, making it clear that we refer only to varieties that are actually suitable for cultivation within a region.

1.13 I126-127. then it would be valuable to show the standard deviations or some measurement of error around those means x-values.

We agree. This plot was changed as response to comment 1.2 and in the new plot we also added the confidence interval around the points.

1.14 I154. please state what are those benefits

We rephrased this sentence to be more specific.

1.15 Fig 4b is rather confusing and hinders relevant information on the uncertainty or error around adaptive capacity estimates relative to each dimension

This part of the figure was removed in the new version of the manuscript.

1.16 I264. Why the choice of reference periods? It would be useful to know what degree of warming these periods represent (with respect to pre-industrial temperatures) to make them comparable with previous studies.

Thank you for this suggestion. We added a more detailed description on the selection of reference periods and the degree of future warming.

1.17 I268. How do you justify the choice of these specific set of models? Please make whatever the justification is, explicit.

We added a description on why these models were selected for our analysis.

1.18 I275. How is model uncertainty dealt with? Since as authors recognise uncertainty is inherent to our inability to exactly predict future climate trajectories, such uncertainty should be explicitly incorporated and reported along with the results. For example, authors could add, even if only as appendices, maps of uncertainty showing where model predictions are more uncertainty (e.g., an uncertainty map for exposure, etc.).

Thank you for this suggestion, which was indeed missing from our original manuscript. Because all our bioclimatic indicators ultimately are derivatives of temperature and precipitation, in our new version, we analyze the model spread for temperature and precipitation over different periods and for several scenarios to highlight regions with increased model uncertainty. The detailed results are now included in the supplementary materials (Supplementary Figure 4 and 5). Additionally, we discuss areas with increased model uncertainty in the main text in the exposure section (Lines 95-103).

1.19 I278. Why any change in climate would be negative (please see major comment related to this)? More so, using a reference period for present time that does not reflect historical conditions driving a given GI to become important (i.e., 1980-83 is often considered as the period when climate warming started to manifest more clearly).

Thank you for this suggestions. We revised our methodology in the new version of the article. Please refer to comment 1.2 for our detailed response.

1.20 1286. The 'biogeographic niche' may be problematic (see major comment related to it), more so when referring to a well-delineated geographic region (a GI). Why not using bioclimatic niche or bioclimatic space, instead?

We replaced the term "biogeographic niche" with "bioregional climate range" throughout the manuscript (see also answer to comment 1.2).

1.21 1298. More details on the specific categorization scheme should be provided here without having to go back to Fraga et al. paper.

We expanded this part and now include a more extensive description of the categorization scheme.

Reviewer #2 (Remarks to the Author):

I really like the idea of this study – it is certainly pertinent to consider the vulnerability of European wine regions and would be of great interest to readers of Nature Communications. However, I have some significant concerns about the ways in which the various scores have been combined.

2.1 Firstly, in relation to how exposure and sensitivity were calculated. Exposure is estimated by examining how mean annual temperature and rainfall are expected to change under climate change. However, there is no real justification provided for the selection of these variables, and the default assumption would actually be that cultivars are more sensitive to a range of other bioclimate metrics. Indeed, and confusingly, other more relevant metrics are used to define the biogeographic niche of the species and to assess sensitivity. The blend of different variables is confusing, poorly explained, and unwarranted. A far more logical approach would have been to use a niche-based modelling approach (aka climate envelope model / species distribution model). In this way, the likely impact of climate change could have been properly quantified as opposed to relying on some rather arbitrary combinations of sensitivity and exposure scores.

Thanks for raising this point. The selection of climate change indicators for the exposure and sensitivity assessment is entirely based on the general guidelines defined by the IPCC in their vulnerability framework². As such, the exposure describes the general change in climatic conditions within a region, while sensitivity describes how a given region is affected by this general change in climatic conditions, considering the specific characteristics of the system (i.e., viticulture). Therefore, for the sensitivity assessment we used more specific bioclimatic indices that were developed for viticulture while the exposure assessment is based on general climatic indices that describe the trend of important climatic variables within a region. Similar approaches have already successfully been used in different studies and realms³⁻⁶. To clarify and better justify the applied approach, we have updated the materials and method section and now provide a more in-depth description of the indicators and how they were selected (Lines 308-326). We also include additional references in the introduction, that show the application of the same framework in different sectors (Lines 56-57) and provide a more open discussion on indicator and framework limitations (Lines 121-129).

Regarding the methodological approach, we agree with the reviewer that a niche-based approach would have likely resulted in a more detailed assessment of the impacts of climate change on vine varieties and regions. Unfortunately, the underlying data for such an approach (i.e., phenological data and detailed information on vineyard locations) are extremely limited and only available for a few wine regions across Europe. For instance, currently more than 1000 different varieties are registered in the approximately 1000 European PDO regions considered in our study, while accurate phenological data

and planting locations are only available for a very small fraction of these varieties and regions (less than 1%, see ref⁷). Thus, for most regions we neither know the phenological characteristics of the cultivated varieties nor we know where, geographically speaking, a specific variety is grown. Therefore, in our study, we opted for a more general approach that allowed us to consider and compare all European PDO regions and their varieties using regional climate groupings, which we now call 'bioregional climate range'. In the new version of the manuscript, we expanded our approach by including statistical data on vineyard area per variety (see comment 1.3 for more details). This allowed a much more accurate estimation of the climate range per variety and thus a better quantification of climate change impacts per variety. Despite the limitations of our approach, we therefore still believe that our study is the first to allow a comprehensive, large-scale assessment of climate change vulnerability across Europe and provides valuable input for the wine sector in facing climate change.

2.2 Second, in the calculation of adaptive capacity, 15 indicators are calculated, and after rescaling to give them a consistent range, they are then just given equal weighting to calculate a total adaptive capacity. Again, however, this seems rather arbitrary, as one would surely expect some to be of greater significance than others. A better approach would have been to consider more carefully how each could or should be weighted and combine the scores accordingly.

Indeed, indicator-based approaches, such as the one we propose here, strongly depend on the selection and weighting of the single indicators. Hence, we have taken great effort in selecting and harmonizing pertinent European-scale datasets that cover the 5 dimensions (i.e., financial, natural, social...) of the framework we applied as best as possible. Still, we cannot fully exclude that some scale effects may occur at the local level in overestimating or underestimating specific dimensions. However, given the comparative nature of our study, we believe that this is of minor concern.

Similarly, we acknowledge that some indicators might be more relevant to specific regions than others and that an indicator weighting would provide more reliable results at a case study level. However, at the continental scale, indicator weighting, in our view, is problematic, as the characteristics of single regions differ significantly. To further support this assumption, and in line with comment 1.4 of reviewer 1, we conducted a survey on indicator importance among five representative wine regions across Europe (see supplementary materials for details). Results indicate that indicator importance is strongly linked to very specific case study peculiarities and that a general pattern on indicator weighting cannot be derived. This result is also supported by different studies where an equal weighting is proposed (i.e., ref⁸). For the sake of clarity, we now discuss our selection and weighting procedure more openly in lines 418-424.

2.3 Similarly, the calculation of total vulnerability as exposure + sensitivity – adaptive capacity is completely arbitrary. There is nothing in the methods to suggest that these have a common scale that make combining them in this way justifiable. While I can see the attraction of having a simple overall metric, it is simply misleading to present it in this way, and it would be far more appropriate to retain these as separate axes.

We respectfully disagree that the vulnerability assessment is completely arbitrary as it strictly follows the guidelines established by the IPCC². There are several examples from previous studies across different sectors and scales where the same framework was used. Please refer to ref⁶ for a review of several international and global indices that use this framework as well as to lines 56-58 in the manuscript for additional references that use the same framework for other agricultural crops.

Regarding the scale of the indicators, all dimensions have been rescaled between 0 and 1, allowing for direct comparability and the calculation of a final vulnerability score. Additionally, we retained and presented the three dimensions separately (e.g. Figure 3). However, we understand the concern of the

reviewer that these results can be difficult to interpret and, in line also with the comments of reviewer 1, we improved our description of the methods used and the general characteristics of the vulnerability framework we adopted (Lines 301-326). In particular, we emphasized that the results from this framework are purely comparative, i.e., the resulting scores can only be interpreted in comparison to the other PDO regions but do not constitute an independent quantitative result (i.e., rating). For example, a vulnerability level of 1, means the highest vulnerability compared to all other European PDO regions, but has no meaning outside this purely comparative context. We hope we could clarify the reviewer's concern about how to use and interpret our approach and results.

2.4 In short, this paper has some very interesting and noteworthy semi-quantitative findings, that really do not lend themselves to being presented as wholly quantitative in the way the authors have done.

We are sorry that you felt that we are overselling our results. Hence, we have taken effort to better clarify and to openly discuss the approach and the methods applied, and how to interpret our results. In particular, we clarified that the results of the vulnerability framework are comparative instead of quantitative, improved the description and selection of indicators, and expanded the description of the overall rationale of the framework that we adopted.

Reviewer #3 (Remarks to the Author):

Overview and significance of work:

3.1 The authors have used an extensive dataset of biophysical, social and economic factors to assess the vulnerability of current wine growing regions to the effects of climate change. The regions are defined by PDOs (Protected Designation of Origin) in some European countries where the assumption is current viticulture is suited to current climate. Of significance is the area covered, and the combination of the biophysical, social and economic factors to assess vulnerability integrating key components from the latest IPCC report. This has not been previously reported for grapevines (often work has focused on the biophysical suitability), representing a novel and comprehensive multidisciplinary approach. The clustering method used enable the authors to find an integrative way to bring together these factors, and resulted in identifying areas of low, medium and high vulnerability by the metrics they used.

The significance of this work is it is the first objective assessment of vulnerability carried out on a scale of significance (several European countries) for viticulture production in the context of climate change. The conclusion that if PDOs are and remain rigid, then resilience to climate change will be reduced and these systems need to be more flexible is relevant. If the key methodological points were addressed, then this would represent a significant advancement in characterising vulnerability. However, the work does not offer solutions (pathways) to addressing the vulnerability. The advance in the field will come from clearly linking the vulnerability outcomes to pathways in an objective manner (see comments below).

3.2 The text was clear and accessible and previous work adequately cited. In terms of references and comparing the work to established literature, information on vulnerability assessment for other crops or species is missing and should be considered.

Thank you for highlighting this point which was indeed not comprehensive in our original version. We now expanded the introduction section and include additional references to previous works that used vulnerability assessments in other sectors and for other crops (lines 56-58).

- 3.3 References were relevant and supported the points raised. It may be that some of the methods considerations outlined below could be addressed also by inclusion of relevant references.

Please directly refer to our responses to the individual considerations below for the newly added references.

Interpretations and validity:

- 3.4 The data interpretations with regard to low, medium and high vulnerability is clear (aside from some important concerns about the methods outlined below which may influence the outcomes of their vulnerability assessment). However, the interpretation for pathways forward on page 7 currently is not supported by the results. For example, nothing in the Figure 3 or its analysis tells us directly that high vulnerability regions need the pathway of biophysical strategies of vineyard management and grape diversification and relocation to mitigate this vulnerability. This interpretation is not directly possible from the analysis presented so far. To draw such interpretation the authors would need to objectively evidence the relative importance of the biophysical components in determining the vulnerability of high vulnerability areas. The authors would need to develop a metric/ or statistical framework for this to be supported. The same comments could be made for the other two pathways outlined in page 7.

Thank you for raising this valuable point. This was poorly formulated in the original version of the manuscript, as we did not intend to derive adaptation pathways directly from the clustering analysis. Indeed, the derivation of individual adaptation strategies for each cluster would be unfeasible, due to the specificities of single wine PDO regions that are determining their possibilities for adaptation. Instead, the described pathways represent a discussion of the main characteristics found in European PDO regions (based on exposure, sensitivity, and adaptive capacity levels) taking into consideration potential adaptation strategies broadly described in the literature. To make this clear, we largely revised the structure of the corresponding section in the manuscript, clearly separating the discussion on potential future pathways from the one on cluster analysis. Moreover, we now also emphasize that this discussion is derived from a comparison of our results to previous studies (lines 236-241).

- 3.5 Furthermore, if high vulnerability areas are predominantly dependent on these biophysical factors, what is the need to include the adaptive capacity in assessing vulnerability for this situation?

According to the IPCC, all three dimensions (exposure, sensitivity, and adaptive capacity) contribute equally to the vulnerability of wine regions, and in our study, we directly build on this framework. While exposure and sensitivity together determine how climate change will impact a certain region, the adaptive capacity contains information on how regions can adapt to these adverse effects. Each dimension therefore contains an important aspect of climate change vulnerability and all three together determine the overall vulnerability of the wine regions. Exposure and sensitivity, for example, determine which strategies might be necessary in a given region (i.e., stronger climate impacts in general necessitate more extensive and long-term adaptation strategies), while the adaptive capacity is important to identify regions that are able to implement such adaptation or that lack resources and might need external investments or subsidies. To make this clearer, we now explicitly explain this relationship on lines 256-269.

- 3.6 In addition, the suggestion to shift production in space is ok but the manuscript failed to address that higher elevation also mean aspect may differ which could also have a large impact on wine style for retaining current PDO characteristics.

We added a clarification, which can now be found on line 251-253.

Data and methodology:

The framework of the methodology is sound and very clearly explained. However, some method aspects should be reviewed.

- 3.7 The details for the various components may not be the best way of assessment the impact. Exposure is also equally weighted to adaptive capacity in the vulnerability assessment but substantially less detailed. Notably the biophysical metrics of annual mean temperature and precipitation over-simplify the situation for calculating “exposure” relative to what is normally used in viticulture and crop production, a calculation of Growing degree-days and specific precipitation patterns at key times of the year for grape production. It is unclear why these metrics were chosen given the depth of work that already exists using more relevant bioclimatic indices or phenological modelling approaches. A wine base study should be at minimum, operating using Growing degree-days (temperatures relevant to the growing season of grapes) and precipitation during this period. Without doing the analysis, it is unclear whether this would change the vulnerability classifications. Why were heat stress and drought metrics also not considered?

Thanks for raising this point. The selection of climate change indicators is entirely based on the general guidelines defined by the IPCC in their vulnerability framework². As such, the exposure describes the general change in climatic conditions within a region irrespective of the studied crop system, while sensitivity considers the specific characteristics of the system (in our case viticulture) and describes how a given region is affected by the general change in climatic conditions described by the exposure. This is why exposure is based on general climatic indicators (e.g., annual mean temperature), while the sensitivity is based on specific bioclimatic indices for viticulture (Huglin Index, Dryness Index and Cool Night Index). Relevant climatic drivers for viticulture (such as heat stress and drought risk) are therefore assessed for individual varieties during the sensitivity calculation via the bioclimatic indices and not for the exposure dimension. Please also refer to the references on lines 58-60 for examples on how the same framework was used across different scales and sectors. To clarify and better justify the applied approach, we have updated the materials and method section and now provide a more in-depth description of the indicators and how they were selected (Lines 301-326). Moreover, we also included a more detailed description of the importance of the three bioclimatic indices for viticulture on lines 367-369.

- 3.8 This also raises the following question: Why the Huglin index and not phenological models were employed for sensitivity analysis where there is quite big differences already described among cultivars by phenological models.

We agree with the reviewer that phenological models would allow for a more detailed assessment of the impacts of climate change on vine varieties and regions. However, the availability of necessary data, such as phenological and vineyard location information, is extremely limited and only accessible for a small number of European wine regions. In our study, which considers all European PDO regions and their varieties, we therefore took a more general approach and used regional climate groupings, now called ‘bioregional climate ranges’, to derive the climate change sensitivity. In the new version of the manuscript, we expanded our methodology by including statistical data on vineyard area per

variety, which allowed for a more accurate estimation of the climate range, including an improved differentiation of individual cultivars. Additionally, now we include a validation of our approach by comparing our results to climate ranges from previous studies which confirms the robustness of the applied approach (supplementary material lines 81-106). For additional details regarding the new methodological approach please refer to comment 1.3 (reviewer 1) and for a more detailed discussion on limitations regarding phenological models and species distribution models please refer to comment 2.1 (reviewer 2).

- 3.9 While the rationale and results of clustering is justified, the weighting of exposure, sensitivity and adaptive capacity in the vulnerability index need to be justified. How and why did each component get equal weighting especially when adaptive capacity amalgamates many metrics and the other two only a few?

As already stated in our reply to comment 3.1, the choice of the three dimensions and their combination to calculate the vulnerability was motivated by the IPCC framework. In the vulnerability framework, each of these dimensions describes a different aspect of the climate change vulnerability of European wine regions, but there is no indication that one dimension might be more important than another dimension in the description of the framework or in our data. Additionally, the considered European PDO regions show substantial differences in many aspects for all three dimensions. Deriving a single set of weights for the three dimensions that is suitable for all regions is therefore unfeasible and doing so might introduce significant biases for many regions. Instead, weights would need to be specified on a case-by-case basis which is unfortunately not possible at the European scale. This is very similar to the weighting of the single adaptive capacity indicators described in the response to comment 1.4 and 2.2. We therefore opted for a neutral approach and, following the specifications in the original IPCC framework, weighted all three dimensions equally.

- 3.10 The origin of the mechanisation index is unclear. One could argue a high dependency on machinery increases adaptive capacity as it leads to increased mechanisation and reduced labour costs (particularly in settings that may score low for Ageing index and dependency ratio). Description does not clearly reflect the calculation.

Thank you for your comment. The reduction in labor costs can indeed be economically beneficial for viticulture. However, when considering the adaptive capacity to a changing climate, other factors become more relevant. A low mechanization index indicates a viticulture that is less dependent on machinery or capable of farming large vineyard areas with relatively few machines. This suggests that their potential to adapt is higher compared to already highly mechanized PDO regions, despite potential initial investment costs. Another rationale for this indicator standardization is the alignment with existing European Union (EU) climate change mitigation strategies. For instance, the new Common Agricultural Policy (CAP) and the EU Farm to Fork Strategy encourage farmers to reduce fossil fuel consumption and increase resilience by reducing dependency on such fuels. Moreover, the EU's European Green Deal aims to make Europe climate neutral by 2050, setting specific targets for GHG emissions reduction, including those in agriculture. By promoting less mechanized approaches in viticulture, there is a better chance of aligning with EU climate goals and fostering resilience in the face of climate change. The description of the index in the supplementary material was expanded to clarify this point.

Analytical approach:

- 3.11 The strengths of the approach is integrating the three metrics in the vulnerability assessment and clustering results. Note that the supplementary material, link 74- 75 state "87% and 75%

of the regions obtain the same scoring as when using 7 or 8 groups, respectively', which means for example for 75%, that 1/4 of regions get a different score depending on the number of groups in the clusters. This represents substantial differences in the groupings of vulnerability depending on the number of clusters used. Given these differences the methods to date do not present an adequate justification of selecting six groups so further justification of this is required.

We agree with the reviewer that our previous explanation was unclear because we did not show the full analysis that was used to derive the number of clusters. We now expanded our justification for the number of clustering groups and present the full analysis of the clustering-performance across various numbers of groups. For this we refer to three indicators that are commonly used to evaluate the performance of clustering algorithms and are related to the within and between-cluster similarity. By including these new results into the supplementary material, we hope to provide a more comprehensive justification for the selection of clustering groups for our main analysis across a wider range of possible numbers of groups.

3.12 There was also an assumption that any change resulted in negative effect because these changes alter the specific biophysical conditions that are needed to produce a certain GI wine. However, between season variability leads to interannual differences in production. So negativity should be assessed relative to this.

Thank you for this important point. Because this assumption is so critical, in the new version of the manuscript we made a substantial effort to improve the estimation of the reference climate range of the authorized varieties and explicitly included an analysis that shows which varieties and regions might benefit from future climate change. For additional details please refer to comment 1.1 of reviewer 1 that highlighted similar concerns. However, we do not explicitly consider seasonal variability and instead base our analysis on long-term climatic conditions. While seasonal climatic variability is certainly important for many aspects of wine production, the style and character of the wine products that originate from a certain region are mostly determined by its long-term climatic conditions and the related wine making techniques that humans have developed in that specific area over the centuries. Indeed, as it is showed by the description of grape productions areas in the PDO regulatory documents, PDO regions and their wines are directly linked to the long-term climatic conditions of a given area (e.g., annual precipitation). These conditions are therefore central to understand how climate change will impact PDO regions in the future, as changes in long-term climate can strongly alter the type and the characteristics of the products from a region, posing a significant threat to its wine products.

Suggested improvements for revision:

3.13 Most key suggestions are covered under the feedback for 'data and methodology' and 'analytical approach'. To summarize:

the metrics used for exposure and sensitivity should be reassessed and may need testing of alternates (outlined in method section). In particular, yearly average temperature is too broad given the current knowledge of bioclimatic indices and phenological models.

The decision to include six clusters needs better justification as highlighted under the section 5. Analytical approach

The weighting of the difference components in the vulnerability index needs justifying

The link between the vulnerability results and proposed pathways needs to be supported by further analysis.

Thank you for highlighting these important points. We have taken great effort to clarify all the highlighted issues, in particular by improving the description of our approach and framework, by expanding and updating the applied methods and by rewriting several parts of the proposed pathways. For the detailed response to each of these points, please refer to the respective comments.

Further minor suggestions for improvement are:

3.14 The term Europe in the heading is too broad. For example, key countries of Germany, Austria, Switzerland, and Greece are missing. The specifics rather than using the term ‘Europe’ also needs to be addressed in the introduction.

We have difficulties in following this point. Our analysis is based on the wine regions published in ref⁹, which covers all European PDO regions that were registered until 04.11.2021 and includes also wine regions in Germany, Austria, and Greece. The PDO regions include the most prominent and famous wines and regions across Europe and, in most countries, cover more than 90% of the vineyard area (ref⁹ Table 2). We therefore believe that our analysis gives a comprehensive overview of the vulnerability of European viticulture and that the term Europe in the heading is applicable to our study. We clearly specified the focus of our study and the covered regions in the material section on lines 290-293 as well as in the introduction on lines 29-30 and lines 64-66.

3.15 Line 42 ‘substitute grape candidate’ – change to grape to cultivar

Corrected

3.16 Line 45- ‘resilience’ this should be defined in terms of what the authors consider resilience relates to

The definition of resilience in the context of our study can be found on lines 39-40.

3.17 Line 135. It is interesting that the adaptive capacity high where exposure was high. What is the explanation?

Exposure and adaptive capacity both tend to be higher in mountain areas, because of the faster rate of climate change in these areas and the increased availability of natural assets under complex topoclimatic conditions. However, there is no direct link between adaptive capacity and exposure and in general there is a weak correlation between the two (pearson`s $r = 0.01$).

3.18 Figure eight suggested vulnerability is predominantly climate driven, how does using VI enable us to demonstrate this importance of integrating adaptive capacity?

In the new version of the manuscript, we completely changed the structure of this section and rewrote large parts to make it clear how adaptive capacity influences vulnerability and why it is an important factor. Please refer to comment 3.5 for more details.

3.19 More detail is required for lines 302- 3-5 in the methods. “Linking all the appellations within a group that grew a specific variety to that variety’. How was this combined with the biogeographical niche of a GI?

The explanation of the sensitivity calculation was greatly rewritten to better explain the new methodological approach and provide more details on how the individual steps were carried out.

3.20 Lines 307- How was the upper limit defined?

The description of the sensitivity calculation was completely reworked in the new version of the manuscript. The definition of the range, with the upper and lower limits, is explained on lines 377-380.

3.21 Education level= why *2 for educated? Justification for the value of this multiplier?

This was done for weighting purposes. A full education level is in most countries equal to a university degree or an agriculture specific high school diploma while a basic education in farming is in most countries any completed farmer training (https://agridata.ec.europa.eu/Qlik_Downloads/InfoSheetSectorial/infoC24.html).

3.22 Line 227- should be explicit these are in non-grape related works (i.e. the approach is a more universal thing)

We removed the part about the pathways from this sentence to avoid any misunderstanding.

3.23 Figure 2- colours in b and c should match

Figure 2b was removed in the new version of the manuscript in response to comment 1.15.

3.24 Abbreviations in figure 2 should be explained in the figure caption

Corrected

3.25 Significant areas on Sup Fig 1 not included

We have difficulties following the comment. Throughout the manuscript we clearly focus on European PDO regions which is specified on lines 29-30 and lines 64-66 in the introduction and on lines 290-293 in the materials section. The figures in the main text all show the same areas as shown in Supplementary Figure 1, only represented as centroids instead of polygons because of the considerable overlap of individual regions in many cases. There are some vineyards across Europe that lie outside PDO regions, but typically they account only for a very small fraction of the total vineyard area for each country (ref⁹ Table 2). For a more detailed description of the PDO regions in Europe, please refer to ref⁹.

3.26 Table 1 repeats Figure 2. Delete table

In the new version of the manuscript and in response to comment 1.4 this table was expanded with additional information regarding the individual indicators. We therefore decided to keep it in the manuscript.

3.27 Suppl. Figure 2-1 hard to read

Figure size was increased to improve readability.

3.28 Suppl. Table 2-1 not clear

This table was removed in the new version of the manuscript.

Reviewer #4 (Remarks to the Author):

This is a very welcome and well-written piece dealing with a vulnerability assessment of European wine regions, under the EU GI system, in relation to climate change to enhance the understanding of which regions are threatened the most by climate change and support the development of potential adaptation pathways to strengthen their resilience.

The authors make this assessment based on the registered wine GIs (PDO/PGI) system and respective regions. The authors specifically state: “Vulnerability is driven by the rigidity of the GI system, which restricts the exploitation of suitable bioclimatic conditions and existing grape cultivar diversity, as well as contextual deficiencies, such as limited socioeconomic resources”.

To assess the climate change vulnerability, the authors adapted the framework developed by the Intergovernmental Panel on Climate Change (IPCC) in which vulnerability is assessed as a function of exposure, sensitivity, and adaptive capacity. To operationalize the assessment the authors defined: (i) exposure as the expected changes in climatic conditions, including temperature and precipitation, (ii) sensitivity as the degree to which a system is affected by climate related stimuli, based on the biogeographical niche of each wine region, and (iii) adaptive capacity as how well a wine region can adapt to changing climate conditions, considering five distinct dimensions (financial, natural, physical, social, and human).

4.1 It is worth noting that the GI system (PDO/PGI) allows for amendments to the product specifications of the protected GIs for wine as specified in the GI legislation the authors add (reference 21). So, the EU GI framework allows for amendments. In fact, the current GI legislation (i.e., food products) is also being amended to cope with sustainability issues, is there also any ongoing modification to the EU (PDO/PGI) regulation for wines? Authors such as Cupri 2020, Edelmann et al. 2020, Marescotti et al. 2020, Quiñones-Ruiz et al. 2018, Teil 2020 (for wine), have analyzed adaptations for safeguarded GIs, mostly involving food products (i.e., fruit and vegetables, cheese), in which producers have to justify why they modify the “rules of the game”, namely the product specifications. Some of the amendments are also related to climate change. It would be worth to specifically show (approved) amendments for wine GIs in general and specifically those dealing i.e., with changing climate conditions, increasing/decreasing geographical area, ... to also observe how producers are formally justifying and coping with external (or internal) factors to their production systems and discuss them with the very interesting findings the authors provide at regional level (1174 wine regions? Or wine GIs?).

Thank you for this important point. We agree with the reviewer that it would be interesting to analyze the amendments of wine GIs in more detail showing how certain changes in legislation were justified and if they were related to climate change. However, there are currently around 400 approved amendments for wine GIs in Europe, and in many cases there is no straightforward and consistent way to identify whether the amendments are related to climate change or to other reasons because there are often multiple motivations behind an amendment. To highlight that amendments to GI regulations are nonetheless possible and occur, we have now reported some more examples in the revised manuscript where climate change was listed as an explicit motivation for changes in the production rules (e.g., lines 196-199 and lines 246-248).

4.2 The authors state: “we assessed the climate change vulnerability of 1174 wine regions in Europe by explicitly considering their biophysical and socioeconomic characteristics and their regulatory specifications”. “Our results provide the first overview of the vulnerability levels of traditional wine products in Europe combined with adaptation pathways that provide critical information to build a more climate-resilient GI system.” Just for clarification, do you specifically talk about 1174 wine regions or wine GIs based in European regions? how many GI wines are you covering in case you deal with 1174 wine regions? in case this data is available and differentiation possible. Does the analysis make a differentiation between PDO and PGI wines or include a differentiation concerning the adaptation pathway and certain trend based on the GI schemes (PDO/PGI)? Do the adaptation strategies showed in the very interesting grouping of pathways specifically deal with already approved amendments just to be precise with the accepted

modifications to the production rules and the wording used related to the EU regulation? This is just to further understand the rigidity the authors aim to show.

In our manuscript, we used the term 'wine region' as a synonym for 'wine GI / PDO' because in Europe they largely overlap (i.e., most of the established wine regions have at least one, sometimes even more, registered GI). In our study, we focused intentionally on the PDO regions only (and not also on PGI) because PDO have the strongest link to the territory and all grapes must be sourced exclusively from the PDO production area. This strong link also helped us in the development of location-specific vulnerability pathways because we knew where the vines and grapes come from and what climate they may experience in the future. To clarify this, we specified it on lines 291-295 in the revised manuscript.

We also provide some examples of already approved amendments in relation to our pathways on lines 196-199 and lines 246-248 (please refer to comment 4.1 for our detailed response).

4.3 Perhaps for sake of clarity, please stick to the GI system, since this is the system the authors deal with, and avoid the term appellation (which is usually used for the national (GI) French system, unless you talk about French appellations) and perhaps briefly state the difference between PDO and PGI to show the most rigid scheme.

Thank you for this valuable suggestion. We removed the term 'appellation' from the entire manuscript and replaced it with GI or wine region, where appropriate. Additionally, we explained the difference between PDO and PGI more clearly on lines 293-295.

4.4 It was not possible to open the links below, please kindly update them if likely:

<https://commission.europa.eu/food-farming-408fisheries/food-safety-and-quality/certification/quality-labels/geographical-indications-409register/details/EUGI00000000861>

<https://commission.europa.eu/food-farming-411fisheries/food-safety-and-quality/certification/quality-labels/geographical-indications-412register/details/EUGI00000002663>

<https://commission.europa.eu/food-farming-481fisheries/food-safety-and-quality/certification/quality-labels/geographical-indications-482register/details/EUGI00000001605>

https://agriculture.ec.europa.eu/farming/geographical-501indications-and-quality-schemes/geographical-indications-food-and-drink/chianti-pdo_en

We update the links and now they lead to the correct website for the different references. Sorry for any inconvenience this may have caused.

4.5 Minor: the paper is very well-written, just double check if only British English is used (i.e., analyzing).

Done.

4.6 Finally, congratulations for this piece!

Thank you.

References

1. Anderson, K. & Nelgen, S. *Database of Regional, National and Global Winegrape Bearing Areas by Variety, 1960 to 2016*. (University of Adelaide, 2020).
2. Haines, A. Climate Change 2001: The Scientific Basis. Contribution of Working Group 1 to the Third Assessment report of the Intergovernmental Panel on Climate Change. JT Houghton, Y Ding, DJ Griggs, M Noguer, PJ van der Winden, X Dai. Cambridge: Cambridge University Press, 2001, pp. 881, £34.95 (HB) ISBN: 0-21-01495-6; £90.00 (HB) ISBN: 0-521-80767-0. *Int J Epidemiol* **32**, 321–321 (2003).
3. Loi, D. T. *et al.* An Assessment of Agricultural Vulnerability in the Context of Global Climate Change: A Case Study in Ha Tinh Province, Vietnam. *Sustainability-basel* **14**, 1282 (2022).
4. Parker, L., Bourgoin, C., Martinez-Valle, A. & Läderach, P. Vulnerability of the agricultural sector to climate change: The development of a pan-tropical Climate Risk Vulnerability Assessment to inform sub-national decision making. *Plos One* **14**, e0213641 (2019).
5. Weis, S. W. M. *et al.* Assessing vulnerability: an integrated approach for mapping adaptive capacity, sensitivity, and exposure. *Climatic Change* **136**, 615–629 (2016).
6. Garschagen, M., Doshi, D., Reith, J. & Hagenlocher, M. Global patterns of disaster and climate risk—an analysis of the consistency of leading index-based assessments and their results. *Climatic Change* **169**, 11 (2021).
7. Morales-Castilla, I. *et al.* Diversity buffers winegrowing regions from climate change losses. *Proc National Acad Sci* **117**, 2864--2869 (2020).
8. Williges, K., Mechler, R., Bowyer, P. & Balkovic, J. Towards an assessment of adaptive capacity of the European agricultural sector to droughts. *Clim Serv* **7**, 47–63 (2017).
9. Candiago, S., Tscholl, S., Bassani, L., Fraga, H. & Egarter, L. V. A geospatial inventory of regulatory information for wine protected designations of origin in Europe. *Sci Data* **9**, 394 (2022).

REVIEWER COMMENTS

Reviewer #1 (Remarks to the Author):

In the revised version of manuscript NCOMMS-22-45849B, the authors have done an excellent job of thoroughly addressing most of my comments and concerns. They have effectively tackled conceptual issues and implemented significant changes and edits that have greatly enhanced the manuscript. Considering its potential basic and applied interest, I believe it is worthy of publication.

I do have a minor remaining comment. In Figure 2, "temperature variability" is still utilized as a criterion to assess the adaptive capacity of regions. However, in the Supplementary Materials, it is referred to as the "availability of climatic niches" (see Line 52). Apart from correcting the discrepancy in the nomenclature of this criterion, I am not entirely convinced that the standard deviation of the monthly mean temperature values accurately captures the concept of climatic niche availability. This kind of metric tends to exhibit consistent bias against coastal regions or grid cells, even though the presence of mountains in these coastal areas could confer a higher availability of climatic niches to a region. In essence, geographic variation would more effectively represent the availability of climatic niches than temporal variation. I realize that calculating geographic variation within point-defined regions might not be feasible. Nonetheless, alternative metrics might better reflect the availability of climatic niches. For instance, an interaction between temperature and topography variation within a region could be more suitable. Also, despite the limitations posed by the provided maps' low resolution and size, I kindly request the authors to meticulously review the maps in Supplementary Figure 5. There seems to be an unusual longitudinal-latitude banding (particularly noticeable in the lower-right panel) that might not correspond to actual climate patterns.

Reviewer #2 (Remarks to the Author):

While I see value in this manuscript, I am disappointed by the extent to which the authors have sought to address my original concerns.

My first concern related to how exposure and sensitivity were calculated. The authors justify their approach by claiming that the selection of climate change indicators for the exposure and sensitivity assessment is entirely based on the general guidelines defined by the IPCC in their vulnerability framework, in which exposure describes the general change in climatic conditions within a region, while sensitivity describes how a given region is affected by this general change in climatic conditions. The report cited in defence of this argument does not include a vulnerability framework. More generally, the IPCC defines exposure as the nature and degree to which a system is exposed to significant climatic variations and sensitivity is as the degree to which a system is affected, either adversely or beneficially, by climate-related stimuli. Thus, the IPCC definitions do not in themselves justify the approach used. They do not imply that exposure should be examined using mean annual temperature and rainfall, whereas sensitivity using range of other bioclimate metrics. The correct approach would be to quantify vulnerability in terms of changes in meaningful bioclimate metrics and to quantify sensitivity in terms of how sensitive grape varieties per unit change in these metrics. I.e. the same climate metrics should be used.

My second concern related to the calculation of adaptive capacity in which 15 indicators were calculated, and after rescaling to give them a consistent range, they were given equal weighting to calculate a total adaptive capacity. The authors justify their approach on the basis that they have taken great effort to select and harmonized and pertinent European-scale datasets that cover the 5 of the framework applied and conclude that while they cannot rule out the possibility that there may be some scale effects at the local level, this is of minor concern given the comparative nature of our study. This is simply untrue. Their results are wholly contingent on weightings applied to each

indicator and there is no reason at all to suppose that they are of equal significance.

My third concern related to the calculation of total vulnerability as exposure + sensitivity – adaptive capacity. The authors state that this is common practise and refer to several papers that approach. I have no issue with the use of exposure, sensitivity and adaptive capacity being used to give an indication of vulnerability. My issue is that it can be calculated simply by summing exposure and sensitivity and subtracting adaptive capacity. The papers given in evidence to refute my concerns do not use this approach. For example, Schilling et al maintain the measures on separate axes as I had suggested. Omerkhil do combine the three elements into one metric, but using an entirely different approach, which is equally rather poorly justified, but either way does not serve to justify the approach used in the present manuscript. In a standard risk assessment exposure and sensitivity are multiplied (together with hazard) to quantify total risk. So it is certainly not standard practise to sum them.

In short, my original concerns still stand.

Reviewer #3 (Remarks to the Author):

The authors have made considerable changes to address points raised in the first review. Overall, the authors revisions have greatly improved the manuscript and addressed many concerns raised from the first review step. The reply here only outlines points that still need addressing.

The mains concerns are related to the sensitivity component focused on variety choice and comments from section 3.8 in the first review (Reviewer #3). If this is addressed clearly, then the work will be of significance for the field.

Firstly, the study has only focused on one attribute important for sensitivity, variety choice. The authors need to explain why variety choice is the one parameter they considered important for the focus of their sensitivity analysis. The authors have not clearly justified why only this, and why other potentially important attributes, for example, rootstocks or training and trellis systems are not considered. Addressing these points would help better define the boundaries of the study plus provide some discussion points on other factors that may need to be considered with regard to sensitivity in future research.

Secondly, using bioclimatic indices to determine varietal suitability is methodologically limited and no longer the best practice in the context of the modelling approaches that now exist to address this. To properly address varietal change, phenology modelling and responses to temperature and precipitation relative to phenological stages is essential. While the authors state a 'lack of phenological characteristics' to do this, there are

1) studies e.g. Morales Castilla et al., (2020), Parker et al., (2013, 2020) that have already used such records successfully with phenology models to assess varietal sustainability in the context of climate change and

2) phenology models from such studies (or others) that can simulate variety suitability in the same method framework as the currently presented index analysis, which don't depend on using observations.

This point requires further revision. The indices so far used, characterize climate conditions rather than the grape response to climate. The comparison in the supplementary methods is against other climate indices (Jones, Fraga) focusing on climate ranges. These climate ranges are not necessarily based on plant response to temperature, rather a proxy measure of where grapes have been grown historically. While these may be interrelated, it is an assumption to say this represents the optimal growing

conditions for a variety (e.g. some varieties may be well suited elsewhere there has just been no history of growing them there). Using the indices to create the thresholds potentially introduces error for projections that then play a role in all subsequent analysis.

Therefore, the authors either need to

1) Avoid over interpreting their bioclimate index results as a measure of variety suitability (and sensitivity in their framework). This will require some considerable reworking of interpretation of what the indices show and highlighting the uncertainty of using these in the projections. The article will need to focus more on how the overall vulnerability assessment is still relevant based on climate with less dependency on the varietal-bioclimate indices relationship.

2) Prove that the bioclimatic index output does indeed correspond to plant development and characterizing varietal suitability. i.e. that current indices reproduce similar outcomes to phenology modelling (not other bioclimate indices) using a few examples of different varieties where they can get data. This was suggested by Reviewer 1 in the first review, and the response from the authors did not yet adequately address it.

Other comments regarding feedback/changes since the first review:

Point 3.9 raised in the first review regarding the equal weighting for all components of the vulnerability index is well justified by the response of the authors. However, there should be a critical evaluation in the discussion within the paper about whether equal weighting is appropriate for the three parts of the Vulnerability index/ and if this is an area that needs further development going forward and the reasons why.

Other specific corrections (line references are for the tracked changes document that was provided):

Line 92 "0.7" Please give brief context when reporting the results as to what this value means (as its only explained in the methods).

Line 285-287 Section is too light on adaptation techniques- rootstocks, irrigation, cover crops etc. all missing.

Line 352 Not correct, many are based on a range of permitted grape varieties, not single varieties

Line 384-388 Highlights that these indices are about changes in climate conditions for characteristics of the viticulture system, not for varietal change

Line 448 There is a big assumption that the GIs where varieties are currently cultivated are the most or only appropriate climate conditions for those varieties. The authors should address that there may be varieties currently not grown in regions that are still suitable to the climate and this is not captured by their approach (i.e., just because a variety is not in a region, does not necessarily mean its not good there now).

Line 465- 467 This description assumes that planting area reflects better suitability by using this weighting. This assumption is risky, and given it is core to the weighting outlined, could lead to bias in results. Authors need to address this issue.

Line 493 "alcohol content" grapes have sugar or potential alcohol but not alcohol itself. Please change.

Line 515-516 It is not clear if the five experts are for each region (the many regions that have been analysed)? How many experts in total?

Reviewer #4 (Remarks to the Author):

I agree with the revisions proposed by the authors.

REVIEWER COMMENTS

Reviewer #1 (Remarks to the Author):

1. In the revised version of manuscript NCOMMS-22-45849B, the authors have done an excellent job of thoroughly addressing most of my comments and concerns. They have effectively tackled conceptual issues and implemented significant changes and edits that have greatly enhanced the manuscript. Considering its potential basic and applied interest, I believe it is worthy of publication.

Thank you for your comment. We are happy to read that you see merit in the modifications that we have made.

2. I do have a minor remaining comment. In Figure 2, "temperature variability" is still utilized as a criterion to assess the adaptive capacity of regions. However, in the Supplementary Materials, it is referred to as the "availability of climatic niches" (see Line 52). Apart from correcting the discrepancy in the nomenclature of this criterion, I am not entirely convinced that the standard deviation of the monthly mean temperature values accurately captures the concept of climatic niche availability. This kind of metric tends to exhibit consistent bias against coastal regions or grid cells, even though the presence of mountains in these coastal areas could confer a higher availability of climatic niches to a region. In essence, geographic variation would more effectively represent the availability of climatic niches than temporal variation. I realize that calculating geographic variation within point-defined regions might not be feasible. Nonetheless, alternative metrics might better reflect the availability of climatic niches. For instance, an interaction between temperature and topography variation within a region could be more suitable.

Thank you for pointing this out. We corrected the discrepancy in the nomenclature by updating Figure 2 in the main text. Moreover, we updated the description of the calculation method, both in Table 1 and in the Supplementary Methods to make it clearer how this indicator was calculated. Indeed, the indicator already contains the geographic variation in thermal conditions, as suggested by the reviewer, and not the temporal variation. Geographic variation was calculated based on the regional boundaries shown in Supplementary Figure 1.

3. Also, despite the limitations posed by the provided maps' low resolution and size, I kindly request the authors to meticulously review the maps in Supplementary Figure 5. There seems to be an unusual longitudinal-latitude banding (particularly noticeable in the lower-right panel) that might not correspond to actual climate patterns.

The maps in Supplementary Figure 5 shows the standard deviation of annual mean temperature between the 5 GCMs considered in the present study. The longitudinal-

latitudinal pattern corresponds to the original ~100km grid cells of the GCMs retrieved from refs^{1,2}. The banding that the reviewer refers to is an artifact from the original climate models and we are therefore not able to remove it.

Reviewer #2 (Remarks to the Author):

While I see value in this manuscript, I am disappointed by the extent to which the authors have sought to address my original concerns.

We are sorry that we were not able to resolve your original concerns. In the new version of the manuscript, we have taken great effort to clarify any remaining issues and to clearly explain the methodological steps that we adopted. Please find our detailed responses to each comment below.

4. My first concern related to how exposure and sensitivity were calculated. The authors justify their approach by claiming that the selection of climate change indicators for the exposure and sensitivity assessment is entirely based on the general guidelines defined by the IPCC in their vulnerability framework, in which exposure describes the general change in climatic conditions within a region, while sensitivity describes how a given region is affected by this general change in climatic conditions. The report cited in defence of this argument does not include a vulnerability framework. More generally, the IPCC defines exposure as the nature and degree to which a system is exposed to significant climatic variations and sensitivity is as the degree to which a system is affected, either adversely or beneficially, by climate-related stimuli. Thus, the IPCC definitions do not in themselves justify the approach used. They do not imply that exposure should be examined using mean annual temperature and rainfall, whereas sensitivity using range of other bioclimate metrics. The correct approach would be to quantify vulnerability in terms of changes in meaningful bioclimate metrics and to quantify sensitivity in terms of how sensitive grape varieties per unit change in these metrics. I.e. the same climate metrics should be used.

We agree that the IPCC definition is rather broad and not limited to the application of a specific approach. In fact, in our study the reasons for choosing general climatic indicators for the exposure dimension were mainly guided by (i) the understanding of exposure as an independent climatic dimension, irrespective from its strict viticultural use, (ii) the widespread use of annual means also in comparable studies in the literature, and (iii) the intention to keep the two dimensions (i.e., exposure and sensitivity) clearly separated by not using the same metric.

At the same time, however, we see the benefits that a consistent use of bioclimatic indices would bring to the study, allowing a better reflection of the growing conditions for the vines and a direct comparison between the two dimensions. Thus, in the revised version of the manuscript, we follow your suggestion and have adopted the same bioclimatic indicators that are used for the calculation of the sensitivity also for the exposure dimension. Consequently, we also adapted the description of the results on lines 90-95, Figure 1a) and c) and the Methods section on lines 356-361.

5. My second concern related to the calculation of adaptive capacity in which 15 indicators were calculated, and after rescaling to give them a consistent range, they were given equal weighting to calculate a total adaptive capacity. The authors justify their approach on the basis that they have taken great effort to select and harmonized and pertinent European-scale datasets that cover the 5 of the framework applied and conclude that while they cannot rule out the possibility that there may be some scale effects at the local level, this is of minor concern given the comparative nature of our study. This is simply untrue. Their results are wholly contingent on weightings applied to each indicator and there is no reason at all to suppose that they are of equal significance.

We fully understand the concerns raised by the reviewer and acknowledge that the weighting of the indicators can indeed have a significant impact on the results of the assessment. However, we argue that given the objective and broad scope of the study and the lack of region-specific data for each of the more than 1000 PDOs across Europe, the approach we have adopted is justified.

In our study, the rationale for applying uniform weightings across all regions was primarily driven by the specific aim and scale of our analysis rather than the assumption that all regions have similar characteristics and that these characteristics are ultimately equally important. Our intention was to provide a broad, high-level overview of the climate resilience of wine regions that would allow meaningful comparisons to be made in a pan-European context. While we recognize that there are local and individual differences between wine regions, and that uniform weightings can never capture all these nuances, we believe that uniform weights are a necessary compromise to facilitate these inter-regional comparisons. To emphasize this point, we have consulted an independent, international panel of experts to show that the weighting of an indicator is highly dependent on the specific circumstances of each region and the individual backgrounds of the experts assigning the weights (see Supplementary Methods). For example, the financial dimension might be the most important factor to face climate change for one expert, while for another expert with a different background, the natural characteristics may be equally or even more important. This shows that adaptive capacity as a concept is inherently multifaceted and context dependent. Introducing weights without a representative panel of experts for each wine region to define region-specific weights would thus likely result in biased outcomes. We therefore opted for an equal weighting as the most neutral approach.

Given the concerns raised by the reviewers, the following changes have been implemented in the revised version of the manuscript: i) we consulted a panel of international experts to clearly show the heterogeneity and variability of indicator importance among individual regions that make clear that assigning weights in comparative studies is not expedient (Supplementary Methods), ii) we expanded the results section, specifically highlighting the limitations of our approach and potential directions for future research (lines 192-198), and

iii) we modified the methods section to clearly explain our choice of equal weights for the adaptive capacity indicators (lines 447-455).

6. My third concern related to the calculation of total vulnerability as exposure + sensitivity – adaptive capacity. The authors state that this is common practise and refer to several papers that approach. I have no issue with the use of exposure, sensitivity and adaptive capacity being used to give an indication of vulnerability. My issue is that it can be calculated simply by summing exposure and sensitivity and subtracting adaptive capacity. The papers given in evidence to refute my concerns do not use this approach. For example, Schilling et al maintain the measures on separate axes as I had suggested. Omerkhil do combine the three elements into one metric, but using an entirely different approach, which is equally rather poorly justified, but either way does not serve to justify the approach used in the present manuscript. In a standard risk assessment exposure and sensitivity are multiplied (together with hazard) to quantify total risk. So it is certainly not standard practise to sum them.

Thank you for pointing this out with greater detail. We realize that the proposed approach could lead to confusion and misinterpretation among the readers with regards to the vulnerability assessment. In the revised version of the manuscript, we therefore refrain from calculating sums and instead simplified the process of indicating the vulnerability score by clearly separating the three components exposure, sensitivity, and adaptive capacity. In this way, we hope to make it very clear, how each component contributes to the overall vulnerability of a region and that each of them describes a distinct characteristic of the regions. To avoid any further confusion on this issue, we completely revised the methodology to combine the three dimensions (explained in the Methods part on lines 463-474) and consequently the visualization in figure 3, which now clearly shows how each of the three dimensions influences the vulnerability of a region and how the different vulnerability levels are characterized. In line with these changes, we also modified the attached files in the external repositories. Moreover, in the description and discussion of the results, we placed stronger emphasis on each individual dimension to clearly show how they influence the impacts and future prospects of PDO regions in the face of climate change, rather than focusing on overall vulnerability.

In short, my original concerns still stand.

Reviewer #3 (Remarks to the Author):

7. The authors have made considerable changes to address points raised in the first review. Overall, the authors revisions have greatly improved the manuscript and addressed many concerns raised from the first review step. The reply here only outlines points that still need addressing.

The mains concerns are related to the sensitivity component focused on variety choice and comments from section 3.8 in the first review (Reviewer #3). If this is addressed clearly, then the work will be of significance for the field.

Thank you for these encouraging words. Please find below our detailed responses and changes to the remaining points.

8. Firstly, the study has only focused on one attribute important for sensitivity, variety choice. The authors need to explain why variety choice is the one parameter they considered important for the focus of their sensitivity analysis. The authors have not clearly justified why only this, and why other potentially important attributes, for example, rootstocks or training and trellis systems are not considered. Addressing these points would help better define the boundaries of the study plus provide some discussion points on other factors that may need to be considered with regard to sensitivity in future research.

This is indeed an important point that was missing in our previous version. We now address this comment by expanding the discussion on the limitations of the sensitivity dimension. More specifically, we now explicitly mention that there are also other parameters that may influence the sensitivity to CC, but the related data is unfortunately only rarely available. The reason for this is partly rooted in the way the regulatory documents of the PDOs were conceived, which were designed to protect the names of specific products and to promote their unique characteristics, for which the consideration of the associated varieties is the most important information. In many cases, they therefore include no or very limited details about other parameters, such as the rootstock type. We have now integrated this on lines 137-143 and hope to have addressed this concern in an appropriate way.

9. Secondly, using bioclimatic indices to determine varietal suitability is methodologically limited and no longer the best practice in the context of the modelling approaches that now exist to address this. To properly address varietal change, phenology modelling and responses to temperature and precipitation relative to phenological stages is essential. While the authors state a 'lack of phenological characteristics' to do this, there are

- 1) studies e.g. Morales Castilla et al., (2020), Parker et al., (2013, 2020) that have already used such records successfully with phenology models to assess varietal sustainability in the context of climate change and
- 2) phenology models from such studies (or others) that can simulate variety suitability in the same method framework as the currently presented index analysis, which don't depend on using observations.

This point requires further revision. The indices so far used, characterize climate conditions rather than the grape response to climate. The comparison in the supplementary methods is against other climate indices (Jones, Fraga) focusing on climate ranges. These climate ranges are not necessarily based on plant response to temperature, rather a proxy measure of where grapes have been grown historically. While these may be interrelated, it is an assumption to say this represents the

optimal growing conditions for a variety (e.g. some varieties may be well suited elsewhere there has just been no history of growing them there). Using the indices to create the thresholds potentially introduces error for projections that then play a role in all subsequent analysis.

Therefore, the authors either need to

- 1) Avoid over interpreting their bioclimate index results as a measure of variety suitability (and sensitivity in their framework). This will require some considerate reworking of interpretation of what the indices show and highlighting the uncertainty of using these in the projections. The article will need to focus more on how the overall vulnerability assessment is still relevant based on climate with less dependency on the varietal-bioclimate indices relationship.
- 2) Prove that the bioclimatic index output does indeed correspond to plant development and characterizing varietal suitability. i.e. that current indices reproduce similar outcomes to phenology modelling (not other bioclimate indices) using a few examples of different varieties where they can get data. This was suggested by Reviewer 1 in the first review, and the response from the authors did not yet adequately address it.

Thank you for this important point. We totally agree with the reviewer that phenological models provide a more accurate tool to estimate varietal suitability by assessing changes in climatic conditions at certain phenological stages. However, we still argue that such an approach is currently not feasible at the European scale when considering the full diversity of different varieties across European PDO regions. While the studies mentioned by the reviewer (i.e., Parker and Morales Castilla) present phenological models for a range of varieties that can be used without requiring additional observations, they do so only for a very limited number of varieties compared to the number of varieties cultivated in European PDO regions. For instance, Morales Castilla et al., (2020) in their study consider 11 different varieties and Parker et al., (2013, 2020) include between 65 and 105 varieties. This covers less than 10% of the over 1000 varieties that are currently registered in European PDO regions. Applying their models therefore would mean neglecting 90% of the varietal diversity currently present in European PDO regions. Additionally, their models were developed based on multi-year time-series of georeferenced phenological observations, which are not available for most of the varieties across European PDOs.

Secondly, and more importantly, the purpose of the sensitivity dimension and the associated bioregional climate ranges is not to assess varietal suitability. Rather, our goal was to assess the historic climatic conditions under which each variety has been traditionally cultivated. This is a critical factor for wine PDO regions, because their reputation, and the very reason for which they are protected, lies in their capacity to produce typical wine products developed under the historic climatic conditions of a specific area. Any deviation from these conditions may therefore cause changes in the characteristics of the wines and pose significant challenges to the PDOs. Our bioregional climate ranges are therefore a proxy for the climatic

ranges under which these traditional products can be produced and not necessarily for varietal suitability. This was explained poorly in our previous version of the manuscript and has now been revised and, hopefully, clarified (lines 380-386, lines 390-392).

Other comments regarding feedback/changes since the first review:

10. Point 3.9 raised in the first review regarding the equal weighting for all components of the vulnerability index is well justified by the response of the authors. However, there should be a critical evaluation in the discussion within the paper about whether equal weighting is appropriate for the three parts of the Vulnerability index/ and if this is an area that needs further development going forward and the reasons why.

Thank you for this point. We added a critical evaluation, in the form of an outlook for future studies on lines 248-255.

Other specific corrections (line references are for the tracked changes document that was provided):

11. Line 92 "0.7" Please give brief context when reporting the results as to what this value means (as its only explained in the methods).

We added an explanation on lines 86-89.

12. Line 285-287 Section is too light on adaptation techniques- rootstocks, irrigation, cover crops etc. all missing.

We expanded the selection of examples and now also include the strategies mentioned by the reviewer (lines 269-272).

13. Line 352 Not correct, many are based on a range of permitted grape varieties, not single varieties

Corrected.

14. Line 384-388 Highlights that these indices are about changes in climate conditions for characteristics of the viticulture system, not for varietal change

This sentence was changed as a response to comment 4.

15. Line 448 There is a big assumption that the GIs where varieties are currently cultivated are the most or only appropriate climate conditions for those varieties. The authors should address that there may be varieties currently not grown in regions that are still suitable to the climate and this is not captured by their approach (i.e., just because a variety is not in a region, does not necessarily mean its not good there now).

Thank you for this important point. Please refer to our response for comment 9 for a detailed answer to this issue.

16. Line 465- 467 This description assumes that planting area reflects better suitability by using this weighting. This assumption is risky, and given it is core to the weighting outlined, could lead to bias in results. Authors need to address this issue.

This is an important point that was addressed also in our response to comment 9. In short, the sensitivity indicator does not reflect varietal suitability but describes the historic growing conditions of each variety, which determine the traditional style and character of the wine products that the PDO regions are known for. We assumed that regions with a large planting area for a given variety are more representative of these historic conditions, because in regions with a small planting area this variety might only be cultivated under very specific microclimatic conditions. Regions with a large planting area therefore gain a higher weight. This is a necessary compromise as there is no information on the exact planting location within a region for each variety, to avoid that regions with small planting areas will bias the resulting range.

17. Line 493 "alcohol content" grapes have sugar or potential alcohol but not alcohol itself. Please change.

Corrected

18. Line 515-516 It is not clear if the five experts are for each region (the many regions that have been analysed)? How many experts in total?

Corrected

Reviewer #4 (Remarks to the Author):

I agree with the revisions proposed by the authors.

Literature

1. Karger, D. N. *et al.* Climatologies at high resolution for the earth's land surface areas. *Sci Data* **4**, (2017).

2. Karger, D. N. *et al.* Climatologies at high resolution for the earth's land surface areas. Preprint at <https://www.envidat.ch/dataset/chelsa-climatologies> (2021).

REVIEWER COMMENTS

Reviewer #1 (Remarks to the Author):

In the latest version of Ms. NCOMMS-22-45849C, the authors have successfully addressed my primary concern raised during the last revision, specifically regarding the availability of climatic niches. Regarding my secondary concern, which pertains to an artificial-looking banding in the climate data, the authors attribute it to the original data but it I have not managed to find such banding in the cited original data sources (perhaps it is actually there too), which could perhaps point towards a data processing error. If alternatively, the source data indeed exhibit such banding, it raises concerns about the data's integrity, and consideration should be given to exploring alternative climate data sources. I would kindly request the authors to provide high-resolution maps for the new Figure S4 to facilitate a more detailed examination of any potential banding. If such banding is confirmed and cannot be rectified, cautionary notes should be included in the legend to alert readers to this limitation.

Reviewer #2 (Remarks to the Author):

I have requested that an editorial decision is made as I think the authors and I will have to agree to disagree over the importance of the concerns I raised in previous reviews of this manuscript.

1. Concern regarding the use of the same bioclimatic metrics for measuring vulnerability and adaptability.

The authors have mostly addressed this concern in that they now use the same bioclimate variables to quantify exposure and sensitivity. Exposure is sensibly measured as the degree of change in relevant bioclimate metrics. Sensitivity is inferred from the range of conditions in which a variety was grown historically, and the method is mostly sensible, though I think would underestimate sensitivity of variates to confined to a narrow range of climates within a specific region. However, much as with adaptive capacity, the act of combining the various measures into a single variable is flawed.

2. Concern regarding equal weighting to calculate adaptation capacity.

I don't think this has been addressed and what has been done is not sensible in my view. The authors have compiled 15 indicators of adaptive capacity, all measured on a different scale and indicating different things and have sought to combine them into a single index simply by rescaling them on the range 0-1. The issue isn't about regional specificity. It is that seeking to collapse variables that measure different things and on a non-common scale into a single measure is arbitrary.

3. Concern regarding vulnerability calculation.

It would be better to provide a standard risk assessment , but it won't negate the more fundamental problem that regions might be vulnerable for a range of different reasons, and that the authors are seeking to calculate a single vulnerability measure without proper consideration of how the different facets of vulnerability combine – some will be additive, some multiplicative and non can be safely assumed to have equal effect simply by statistical rescaling the variables.

Reviewer #3 (Remarks to the Author):

The authors have been very thorough in addressing my feedback and I would like to acknowledge the significant contribution that this work will make to the field of research.

Outstanding minor revisions now suggested are:

1. Addressing in text the point of the limitations of the approach of using bioclimatic indices and not phenology models and why it was not possible to use the latter in the study. The authors provided a good justification but did not make any in-text changes to address this methodological point (this was in response to point #9, reviewer 3 in the previous reviewer comments).
2. Check the manuscript for the use of variety and cultivar. Chose one rather than alternate (e.g. see line 113).

REVIEWER COMMENTS

Reviewer #1 (Remarks to the Author):

1. In the latest version of Ms. NCOMMS-22-45849C, the authors have successfully addressed my primary concern raised during the last revision, specifically regarding the availability of climatic niches. Regarding my secondary concern, which pertains to an artificial-looking banding in the climate data, the authors attribute it to the original data but it I have not managed to find such banding in the cited original data sources (perhaps it is actually there too), which could perhaps point towards a data processing error. If alternatively, the source data indeed exhibit such banding, it raises concerns about the data's integrity, and consideration should be given to exploring alternative climate data sources. I would kindly request the authors to provide high-resolution maps for the new Figure S4 to facilitate a more detailed examination of any potential banding. If such banding is confirmed and cannot be rectified, cautionary notes should be included in the legend to alert readers to this limitation.

Thank you for pointing this out. We checked our calculations and repeated the relevant steps manually using different software to make sure the banding mentioned by the reviewer does not stem from a data processing error.

It should be noted that this banding effect only appears when calculating the standard deviation between the individual climate models. This artifact is likely a result of the downscaling procedure and is probably related to the b-spline interpolation applied to the global climate models during the downscaling¹. By calculating the standard deviation between the individual models, the variance between different interpolation results becomes apparent. The banding, however, is neither present in the raw downscaled data of the individual models, nor in our calculated indices. Therefore, we believe that the banding does not compromise the integrity of the original data, which has already been used and validated in multiple studies, for example refs²⁻⁶. To make this clear to the reader and avoid any misunderstandings, we included high-resolution maps of the raw bioclimatic indices in the supplementary file that show the structure and resolution of the underlying climate data (Supplementary Figure 4).

Reviewer #2 (Remarks to the Author):

I have requested that an editorial decision is made as I think the authors and I will have to agree to disagree over the importance of the concerns I raised in previous reviews of this manuscript.

2. Concern regarding the use of the same bioclimatic metrics for measuring vulnerability and adaptability

The authors have mostly addressed this concern in that they now use the same bioclimate variables to quantify exposure and sensitivity. Exposure is sensibly measured as the degree of change in relevant bioclimate metrics. Sensitivity is inferred from the range of conditions in which a variety was grown historically, and the method is mostly sensible, though I think would underestimate sensitivity of variates to confined to a narrow range of climates within a specific region. However, much as with adaptive capacity, the act of combining the various measures into a single variable is flawed.

3. Concern regarding equal weighting to calculate adaptation capacity.

I don't think this has been addressed and what has been done is not sensible in my view. The authors have compiled 15 indicators of adaptive capacity, all measured on a different scale and indicating different things and have sought to combine them into a single index simply by rescaling them on the range 0-1. The issue isn't about regional specificity. It is that seeking to collapse variables that measure different things and on a non-common scale into a single measure is arbitrary.

4. Concern regarding vulnerability calculation.

It would be better to provide a standard risk assessment, but it won't negate the more fundamental problem that regions might be vulnerable for a range of different reasons, and that the authors are seeking to calculate a single vulnerability measure without proper consideration of how the different facets of vulnerability combine – some will be additive, some multiplicative and non can be safely assumed to have equal effect simply by statistical rescaling the variables.

We regret that we were unable to address your concerns. With the newest modifications to our manuscript and the suggestions we received from the editorial team, we hope to better highlight for the reader the limitations in our analysis and avoid any misunderstanding in how to interpret the results.

Reviewer #3 (Remarks to the Author):

The authors have been very thorough in addressing my feedback and I would like to acknowledge the significant contribution that this work will make to the field of research.

Thank you for your feedback. We are happy to read that you see merit in our work.

Outstanding minor revisions now suggested are:

5. Addressing in text the point of the limitations of the approach of using bioclimatic indices and not phenology models and why it was not possible to use the latter in the study. The authors provided a good justification but did not make any in-text changes to address this methodological point (this was in response to point #9, reviewer 3 in the previous reviewer comments).

Thank you for this feedback. We made several changes in the new version of the manuscript to better highlight the limitations of our approach. The limitations regarding the lack of phenological data and why it was necessary to base our analysis on more general bioclimatic indices instead of using phenological models is described on lines 126-137. Additionally, we inserted a new paragraph on lines 305-323, which synthesizes the limitations in our approach and their implications for our findings and for future research.

6. Check the manuscript for the use of variety and cultivar. Chose one rather than alternate (e.g. see line 113).

Thank you for this suggestion. We now use the word "variety" throughout the whole manuscript.

References

1. Karger, D. N. *et al.* Climatologies at high resolution for the earth's land surface areas. *Sci Data* **4**, 170122 (2017).
2. Thuiller, W., Guéguen, M., Renaud, J., Karger, D. N. & Zimmermann, N. E. Uncertainty in ensembles of global biodiversity scenarios. *Nat. Commun.* **10**, 1446 (2019).
3. Bruehlheide, H. *et al.* Global trait–environment relationships of plant communities. *Nat. Ecol. Evol.* **2**, 1906–1917 (2018).
4. Poggio, L. *et al.* SoilGrids 2.0: producing soil information for the globe with quantified spatial uncertainty. *SOIL* **7**, 217–240 (2021).
5. Berdugo, M. *et al.* Global ecosystem thresholds driven by aridity. *Science* **367**, 787–790 (2020).
6. Klink, R. van *et al.* Meta-analysis reveals declines in terrestrial but increases in freshwater insect abundances. *Science* **368**, 417–420 (2020).